# Crown ether decorated silicon photonics for safeguarding against lead poisoning

Luigi Ranno [1,7], Yong Zen Tan[2,7], Chi Siang Ong[2], Xin Guo[3], Khong Nee Koo[4], Xiang Li[3], Wanjun Wang[3], Samuel Serna[1], Chongyang Liu[5], Rusli[3], Callum G. Littlejohns[6], Graham T. Reed [6], Juejun Hu [1], Hong Wang [3] & Jia Xu Brian Sia [1,3] ✉

Lead (Pb$^{2+}$) toxification is a concerning, unaddressed global public health crisis that leads to 1 million deaths annually. Yet, public policies to address this issue have fallen short. This work harnesses the unique abilities of crown ethers, which selectively bind to specific ions. This study demonstrates the synergistic integration of highly-scalable silicon photonics, with crown ether amine conjugation via Fischer esterification in an environmentally-friendly fashion. This realizes an integrated photonic platform that enables the *in-operando*, highly-selective and quantitative detection of various ions. The development dispels the existing notion that Fischer esterification is restricted to organic compounds, facilitating the subsequent amine conjugation for various crown ethers. The presented platform is specifically engineered for selective Pb$^{2+}$ detection, demonstrating a large dynamic detection range, and applicability to field samples. The compatibility of this platform with cost-effective manufacturing indicates the potential for pervasive implementation of the integrated photonic sensor technology to safeguard against societal Pb$^{2+}$ poisoning.

Anthropogenic lead (Pb$^{2+}$) poisoning represents one of the primary public health concerns since antiquity. Pb$^{2+}$ is a cumulative toxicant that leads to multifaceted impact on biological functions[1-5]. Pb$^{2+}$ has the affinity to substitute other bivalent and monovalent cations. For instance, Pb$^{2+}$ can replace Ca$^{2+}$ ions to cross the blood-brain barrier, resulting in neurological deficits[1]. This effect is exacerbated in children due to the ongoing development of their neurological and nervous system[2]. Pb$^{2+}$ is also found to impact cardiac function, causing a reduction in the speed of heart contraction and relaxation[4]. Furthermore, fetal exposure can result in a wide array of risks during pregnancy[3]. The above examples only serve to highlight a non-exhaustive overview of the impact of Pb$^{2+}$ on our society[6]. However,

public actions against lead toxification are disproportional to its impact. It has been estimated that lead service lines still deliver drinking water to about ten million households in the United States[7]. The impact of Pb$^{2+}$ leads to the conclusion that there should be zero tolerance to lead exposure. To that effect, the Environmental Protection Agency (EPA), US has implemented a limit of 15 parts-per-billion (ppb) in drinking water[8]. Lead poisoning is even more pronounced in developing countries, where the World Health Organization (WHO) estimates that of the 240 million people that are overexposed, 99% comes from developing countries[9,10]. Pb$^{2+}$ exposure accounts for more than one million deaths annually, with significant societal and economic costs, specifically in developing countries[9,10]. These facts

[1]Department of Materials Science & Engineering, Massachusetts Institute of Technology, Cambridge, MA, USA. [2]Fingate Technologies Pte Ltd, 8 Cleantech Loop #06-65, 637145 Singapore, Singapore. [3]School of Electrical and Electronic Engineering, Nanyang Technological University, 50 Nanyang Avenue, 639798 Singapore, Singapore. [4]Vulcan Photonics SDN. BHD. D-11-08, Menara Suezcap 1 KL Gateway, No. 2, Jalan Kerinchi, Kampung Kerinchi, 59200 Kuala Lumpur, Malaysia. [5]Temasek Laboratories, Nanyang Technological University, 50 Nanyang Avenue, 637553 Singapore, Singapore. [6]Optoelectronics Research Centre, University of Southampton, Southampton SO17 1BJ, UK. [7]These authors contributed equally: Luigi Ranno, Yong Zen Tan. ✉e-mail: jiaxubrian.sia@ntu.edu.sg

highlight the urgency for the development of technologies that guard against $Pb^{2+}$ toxification.

Contemporary methods for $Pb^{2+}$ detection can be grouped into two primary categories: Inductively Coupled Plasma Mass Spectrometry/Optical Emission Spectroscopy (ICP-MS/OES)[11], and colorimetric test strips[12]. The former represents the state-of-art, but however, suffers from low sample throughput, requiring extensive sample preparation. Furthermore, such systems are dedicated for lab use only, and not viable for *in-operando* analysis[11]. Colorimetric strips, while low-cost, are qualitative and might lack accuracy[12]. The crown ether-decorated silicon photonic (SiP) sensing platform realizes chip-scale integration of these sensor systems. This enables highly quantitative and selective sensing as well as rapid, portable detection capabilities within a fully integrated platform. It is noted that predating the demonstration of this work, there have been significant development in discrete sensors based on square wave anodic stripping voltammetry (SWASV) electrochemistry[13,14], fluorescence[15], colorimetry (quantitative)[16], and more lately, fiber-based technologies[11] with impressive performances. However, discrete sensors require analytical instruments on a separate platform, limiting the scalability of implementation. This is distinct from our photonic platform in view of its compatibility with waveguide-based analytical components[17] within the SiP ecosystem. Supplementary Note 1 provide an overview and discussion of our platform and the abovementioned technologies.

The photonic sensor platform in this work makes use of crown ethers, cyclic polyethers consisting of multiple oxygen atoms, forming a ring structure[18]. These class of compounds were first synthesized by Charles Pederson in the 1960s, who was subsequently awarded the 1967 Nobel prize for this discovery[19]. As a result of the cavity, which arises from the ring structure, crown ethers possess a remarkable ability to selectively bind to certain ions or molecules based on their properties such as size selectivity, charge accommodation, ring geometry, and structure energetic favorability[19-21]. Crown ethers have been demonstrated on electrochemical[22] and fluorescence[23]-based detection schemes as well. However, as previously outlined, scaling these discrete technologies to low-cost large sensor arrays for widespread detection remains a challenge.

In this manuscript, we demonstrate a crown ether functionalized SiP platform. Traditionally, the functionalization of crown ether on silicon involves the use of silylating agents with trisubstituted silyl groups, which are moisture and pH-sensitive, and require stringent process control[24,25]. In the above protocol, the reagents can potentially undergo self-reaction, resulting in agglomeration, which decreases surface uniformity, impacting sensor reproducibility[24,25]. The application of the Fischer esterification protocol[26] to couple carboxylic acid groups with the -OH group on pretreated $SiO_2$ waveguides surfaces, demonstrated in this study, can circumvent the aforementioned problem and produce the uniform amine conjugation of crown ethers on the $SiO_2$-coated waveguide surfaces (see below Fig. 3a). We note that the successful Fischer esterification on $SiO_2$ defies the conventional view that the reaction is applicable to organics only[26]. Toward a broader scope, the Fischer esterification of an inorganic material possessing an -OH group implies the surface-agnostic nature of this process (-OH group), indicating far-reaching technological implications to replace silylation agents in cases where they are used to couple silica/silicon with organic compounds[24,25]. For instance, different crown ethers[27-37], selective to various ions (i.e., $K^{30}$, $Be^{31}$, $Ra^{33}$, $Cs^{28}$), illustrated at the inset of Fig. 1e, can undergo amine conjugation following Fischer esterification on $SiO_2$, greatly broadening the range of applications (i.e., medical[30], electronics manufacturing[31], nuclear[28,33]) that the developed platform can be extended to. As a corollary of complementary metal-oxide-semiconductor (CMOS) fabrication, SiP has proven to be a disruptive integrated photonic technology that enables high-precision mass manufacturing[38]. Through the synergistic integration of both technologies, the resulting platform is engineered to

overcome several unaddressed issues against lead poisoning in society: 1). The successful amine conjugation of crown ethers via Fischer esterification onto aptly designed SiP circuits will enable the selective, ppb-scale detection of $Pb^{2+}$ ions with large dynamic range (1−62000 ppb), improving upon current bulky lab-based systems (ICP-MS/OES)[11] in terms of portability. The photonic sensor enables the detection of $Pb^{2+}$ concentrations below the EPA limit (15 ppb)[8]. Moreover, the large upper bound of detection broadens its applicability to heavy industries such as mining[39], smelting[40], battery manufacturing[40], effluent monitoring[40] where detection at the tens of part-per-millions (ppm) is mandated. 2.) ICP-MS/OES requires a significant lead time from sample collection to results due to complex lab-based sample processing and analysis[11]. Our sensor enables a single measurement to be performed in ~120 s. This is in addition to its ability to maintain high detection accuracies across a pH range of 6 − 8 (limited by Pb solubility at higher pH[41]), encompassing typical environmental conditions[42], without requirements on sample processing. This indicates the capacity for rapid, *in operando* determination of $Pb^{2+}$ concentration on site. 3). The viability of the sensor platform is assessed using field-collected samples (including tap, lake and seawater samples), where no impairment in detection accuracies was observed. This highlights the potential of the sensor to be applied in real life applications. 4). The high-index contrast of silicon against its cladding material ($SiO_2$) enables the design of compact integrated photonic sensors. The realization of the $Pb^{2+}$-selective sensor within the SiP ecosystem implies compatibility with established waveguide-based analytical components (i.e., spectrometer[17]). This enables sensor system integration (sensor and analytics[17]) at the chip-scale. 5). Through established silicon manufacturing, SiP circuits can be mass-produced at low cost[38]. Furthermore, the crown-ether functionalization process is solution-based (reactants dissolved in green solvents, such as water and ethanol), implying that wafer-scale functionalization can be achieved, indicating scalability, with minimal environmental pollution. The ability of the integrated platform to leverage on highly-precise, scalable, and cost-effective fabrication techniques serves to decrease the price of compact and portable $Pb^{2+}$ sensor systems, without compromising performance. This encourages the much-needed proliferation of such $Pb^{2+}$ detection systems in society.

## Concept
### Photonic device design
The $Pb^{2+}$ sensor illustrated in Fig. 1a, is fabricated on the 220 nm silicon-on-insulator platform; the silicon device layer is 220 nm thick. As indicated in Fig. 1b, the sensor consists of 3 primary technology sublayers: SiP, Fischer esterification, and crown ether-based functionalized layer. Fischer esterification lays the ground for the amine conjugation of crown ethers, where the subsequent change in the material index of the crown ether-based functional layer upon detection of $Pb^{2+}$ ions is translated into the optical domain via SiP. The micrograph image of the sensor is shown in Fig. 1c. Slot waveguides are implemented in the sensing region ($H_2O$ cladding in the aqueous phase). As the slot width is comparable to the exponential decay length of the fundamental eigenmode, optical power perpendicular to the high-index contrast interfaces is amplified[43,44]. Essentially, this feature of slot waveguides lends to high surface sensitivity[43,44]. The input lightwave propagates to an asymmetric adiabatic tapered splitter[45] (Fig. 2d), where a larger proportion of the optical power is directed to the sensing path. Following, a 250 μm-long strip-to-slot converter is utilized for the transition of the strip-to-slot optical mode[46] (Fig. 2e). With the exception of the sensing region (Fig. 1a, b), which is exposed to the analyte in the aqueous phase, the entire device is cladded with $SiO_2$. The thickness of the $SiO_2$ cladding is designated to be 2 μm to prevent interaction with the analyte, where >99% of the optical power is confined within the cladding. In the reference arm, slot waveguides with identical dimensions are also implemented. This is to normalize

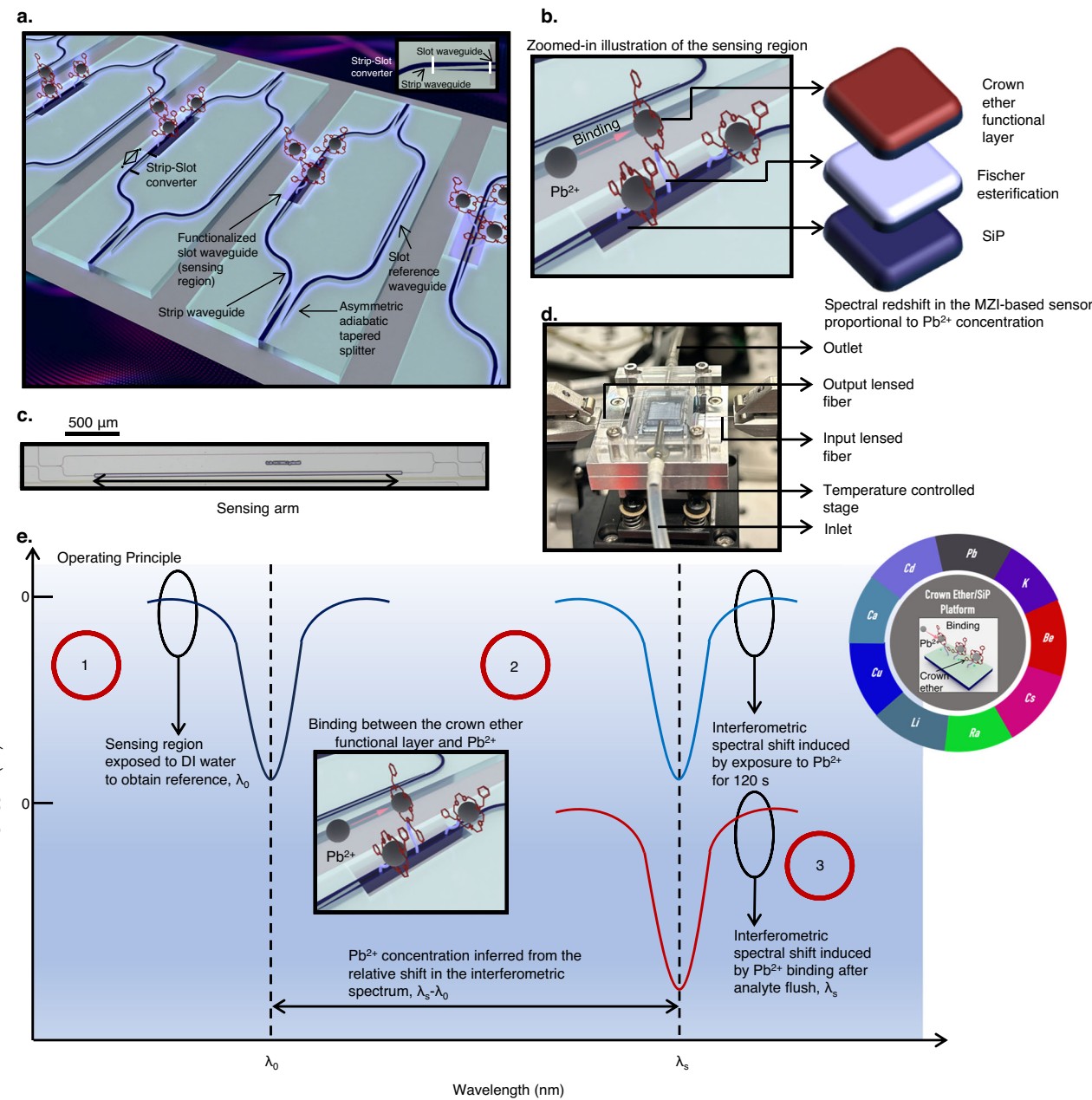

**Fig. 1 | Concept of the Crown Ether/SiP platform for Pb²⁺ ion detection. a** 3-D illustration of the photonic $Pb^{2+}$ ion sensor based on the crown ether-decorated SiP platform. The functionalization performed in the sensing region is indicated. For the sake of clarity, the 20 nm $SiO_2$ deposited on top of the waveguides in the sensing region is not indicated. Information is provided in Fig. 3a. **b** The zoomed-in illustration of the sensing region, where the binding between the $Pb^{2+}$ ions and crown ether functional layer is depicted. The technology sub-layers in the sensing region are labeled: SiP, Fischer esterification, and the crown ether functional layer. **c** Micrograph image of the $Pb^{2+}$ ion sensor, where the sensing arm and scale bar (500 μm) are indicated **d** The $Pb^{2+}$ photonic sensor assembly, consisting of the photonic chip and a microfluidic chamber. **e** Elucidated operating principle of the photonic $Pb^{2+}$ ion sensor. The inset shows the exemplary applications that the ion detection platform can be extended to.

waveguide propagation losses on the sensing and reference arms, and the power ratio of the asymmetrical adiabatic tapered splitters are designed according to $H_2O$ absorption[47] (designed losses) in the sensing region. The lightwave from the sensing and reference arms recombines at the asymmetrical adiabatic tapered splitter, forming the MZI interferometric spectrum. The power ratio of the two splitters is designed to optimize interference fringe visibility[48] (extinction ratio). The operating protocol of the sensor is elucidated in Fig. 1e. First of all, the sensing arm is exposed to deionized (DI) water to obtain the reference resonant wavelength ($\lambda_0$); subsequent wavelength shift is considered in reference to this wavelength.

Then, DI water is flushed from the microfluidic chamber, and an analyte possibly containing $Pb^{2+}$ ions is added and exposed for 120 s. The temporal dependence of the sensor is analyzed through the crown ether functional layer via X-ray Photoelectron Spectroscopy (XPS) where the signal strength rises from 10 s to 120 s and remains stable after (See Supplementary Note 2). To that effect, all the following analyte exposure times are standardized at 120 s. Should the analyte contain $Pb^{2+}$, a resonant shift is induced through the binding of ions to the functionalized surface of the sensing region via surface sensing[44]. However, one should also note that a proportion of the wavelength shift could also be caused by the interaction of the evanescent field

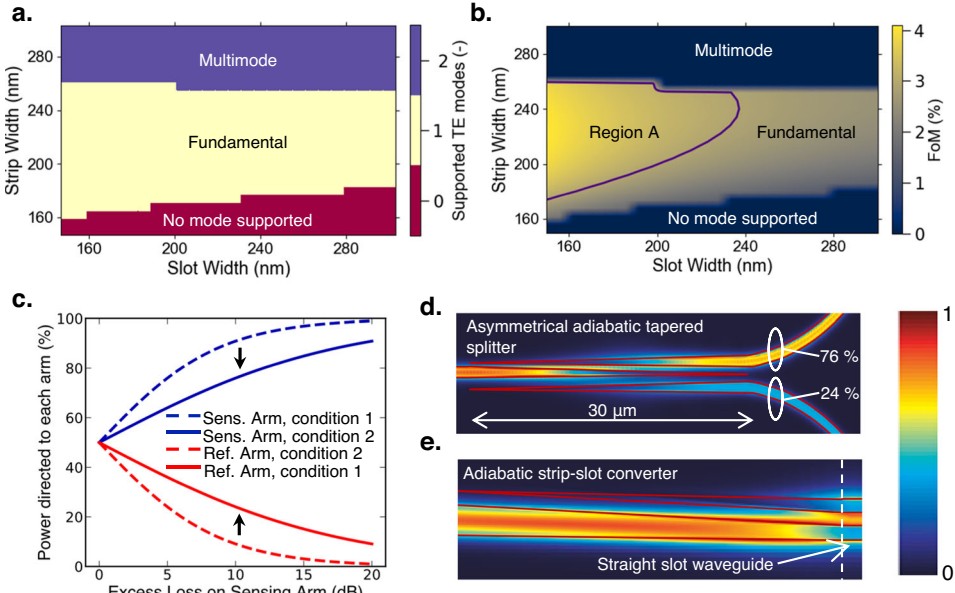

**Fig. 2 | Photonic design of the Pb²⁺ ion sensor. a** Simulation of the number of supported TE optical modes in the slot waveguides as a function of strip and slot width. **b** Sensor surface sensing FoM as a function of strip and slot width. **c** The comparison of two proposed splitting Mach-Zehnder architectures (see Supplementary Note 5) in terms of the power asymmetry required of the splitter;

condition 1 ($S_1 = S_2, S_1' = S_2', S_{1/2}' \neq 0.5$), condition 2 ($S_1' \neq 0.5, S_2' = 0.5$). Top-down electric field distribution of the **d**, asymmetrical adiabatic tapered splitter, and **e**, adiabatic strip-slot converter, where the structures of the components are outlined.

with the other particles/ions/molecules in the analyte[44]. Therefore, this necessitates the subsequent flushing of the analyte from the device via DI water. This results in the retention of Pb²⁺ ions, which are bound to the surface of the sensing region through the crown ether-based functionalized layer, where Fig. 1b provides a further illustration. The complexation of Pb²⁺ ions with the crown ether can be understood through the interaction of the 6s2 electrons and accepting 6p orbitals with the electron lone pairs on the donating oxygen and nitrogen groups on the crown[49,50]. When Pb²⁺ ions are binded to the crown ether-based functional layer, the high electron density of Pb²⁺ increases the polarizability of the functional layer[51,52]. This results in a rise in the relative permittivity of the functional layer. Consequently, the material refractive index increases with higher relative permittivity. Subsequently, due to the higher group index of the sensing waveguide as compared to the reference arm, an increase in the material index of the functional layer will lead to a resonant redshift ($\lambda_s$-$\lambda_0$), attributed to the crown ether-immobilized Pb²⁺ ions in the sensing region. The concentration of the Pb²⁺ ions in the analyte can then be deduced from the relative shift in resonance wavelength from reference ($\lambda_s$-$\lambda_0$), utilizing a calibration curve; the calibration curve indicates ($\lambda_s$-$\lambda_0$) as a function of Pb²⁺ concentration (see below Fig. 5b). We will henceforth refer to the flushing of analyte and the addition of DI water into the microfluidic chamber after ion interaction as analyte flush.

Figure 1d shows the photonic chip, with the polydimethylsiloxane (PDMS) microfluidic channel mounted via a stainless-steel fixture. The microfluidic channel is designed to hold 0.426 ml of analyte, where the solution is filtered through a syringe filter (0.45 μm) prior to sensor exposure. Analyte input and extraction was implemented via the following inlet and outlet tubes. Optical input/output was performed via edge coupling between a lensed fiber with ~3 μm mode field diameter and a silicon coupler that tapers down to 175 nm wide. The abovementioned assembly (see Supplementary Note 3) was mounted on top of a thermoelectric controller (TEC), maintained at 296 K with a thermal drift of lower than 2 mK.

The dimensions of the slot waveguide (Supplementary Note 4) were determined via eigenmode calculations in Fig. 2a, b, where H₂O

cladding surrounds the structure. The parameter space corresponding to the number of transverse electric (TE) modes was performed in Fig. 2a. As the top and bottom media surrounding the waveguide are asymmetrical (BOX on the bottom and H₂O as the cladding), there exists a regime where the fundamental mode is not supported (in red). Conversely, the multi-mode regime of the slot waveguide structure is indicated in blue, where the second-order TE mode can be supported. According to Fig. 2a from left to right, the second-order TE mode emerges when the strip and slot widths are ~260 and ~145 nm, respectively. In addition, the corresponding strip width that excites the second-order TE mode decreases as slot width increases. The parameter space corresponding to single TE mode propagation is highlighted in yellow. For optimization of surface sensitivity, the selection of the optimal strip and slot width, subject to the fundamental TE mode, is dependent on the optical mode confinement on the surface of the sensing region. To that effect, a figure-of-merit (FoM) is defined, that takes into consideration the optical confinement factor within 10 nm about the surface of the slot waveguides, which are cladded with 20 nm of SiO₂ (see Supplementary Note 4).

Computation of the FoM is performed in the parameter space of Fig. 2a and the results are presented in Fig. 2b. The corresponding boundary condition for the number of supported TE modes (Fig. 2a) is replicated in Fig. 2b. As a guide to the eye, the highest value of FoM is indicated by region A. However, we have encountered difficulties in the selective removal of SiO₂ cladding at the sensing region when the slot gap is smaller than 200 nm. As such, strip and slot width of 240 and 240 nm, respectively are implemented to relax process requirements; the selected slot waveguide parameters lie close to the boundary of region A.

H₂O poses significant water absorption at the C-band[47]. Yet, the length of the sensing region increases the surface sensitivity; which forces an inherent tradeoff between the fringe visibility (extinction ratio) and sensitivity[44]. The implementation of asymmetrical splitting in MZIs will serve to overcome the issue. As identical slot waveguide dimensions are implemented on the sensing and reference arms, the primary source of loss difference between the two arms comes from

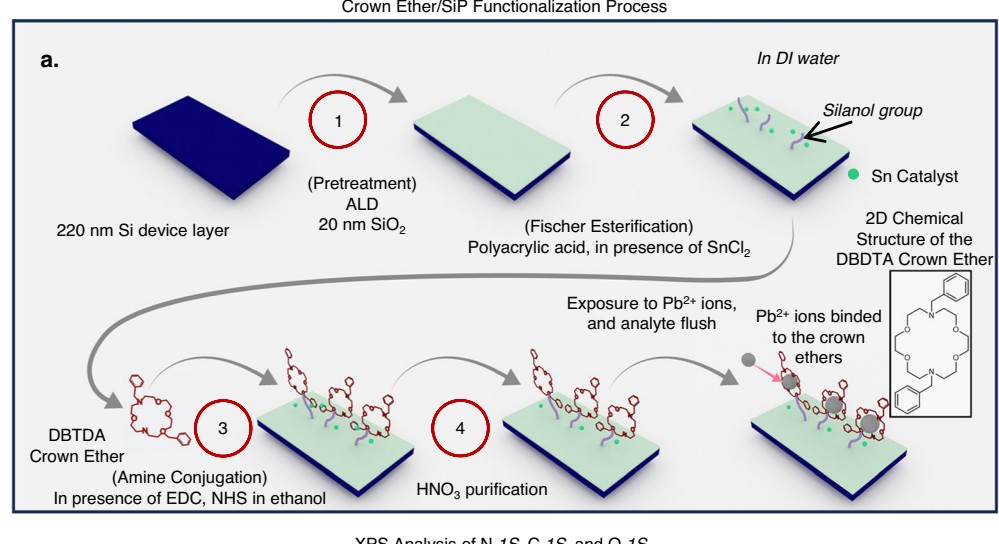

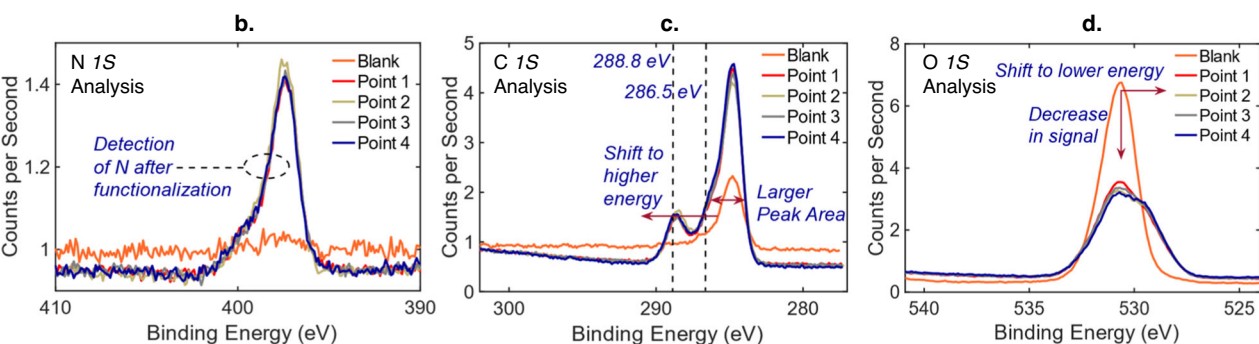

**Fig. 3 | The development of the Crown Ether/SiP functionalization process.**
**a** The developed crown ether/SiP functionalization process, described in four steps; the 2D chemical structure of the DBTDA crown ether is illustrated. XPS narrow spectra analysis of the **b**, N $1S$, **c**, C $1S$, **d**, O $1S$ regions of the photonic chips, before and after functionalization.

water absorption. It can be concluded that the splitting ratios of the MZIs must be co-designed with the length of the sensing arm, hence designed losses. The quantities are related to one another via the following equation, where the derivation is elaborated (see Supplementary Note 5). Designed losses through water absorption is assumed to be the only source of loss in the sensor.

$$\sqrt{S_1}\sqrt{S_2}e^{-\frac{\alpha L}{2}} = \sqrt{S_1'}\sqrt{S_2'} \qquad (1)$$

$S_1, S_1'$ and $S_2, S_2'$ refers to the splitting ratios of the input and output splitters, respectively. Assuming the splitters are lossless, energy conservation dictates that $S_{1/2} = 1 - S_{1/2}'$. $\alpha$ is the loss coefficient due to water absorption, and $L$ refers to the length of the sensing arm. We propose a condition such that $S_1 = S_2$, $S_1' = S_2'$ and $S_{1/2}' \neq 0.5$ (condition 1). In Fig. 2c, we plot the splitting ratios to the sensing and reference arm corresponding to maximum visibility. A comparison to an alternate condition where arbitrary splitting is at the input splitter, and 3-dB splitting is at the output splitter (condition 2, $S_1' \neq 0.5, S_2' = 0.5$) is also indicated in Fig. 2c; see Supplementary Note 5. In comparison, condition 1 reduces the asymmetry that is required of the splitters, alleviating fabrication requirements. As a compromise between the sensor surface sensitivity and the optical measurement setup power budget, our demonstration selected splitting ratios corresponding to sensor arm designed loss of 10 dB: $S_1 = 0.76, S_2 = 0.24$. In regard to the selected slot waveguide dimensions in a water cladding, the waveguide propagation loss due to $H_2O$ absorption is estimated to be 35 dB/cm at

$\lambda = 1.55$ µm. This gives rise to a sensing arm length of $\frac{\text{Designed Loss (dB)}}{\text{Propagation Loss (dB/cm)}} = \frac{10}{35} = 2857.14$ µm.

Asymmetrical adiabatic tapered splitter, which has been developed in our previous work[45], are implemented for arbitrary power splitting. In Fig. 2d, we show the top-down electrical field distribution of a 30 µm-long 76/24% power splitter. The length of the strip-to-slot converter is 250 µm, where low-loss adiabatic conversion from strip to slot mode is facilitated[46]. Similarly, the top-down electric field distribution of the lightwave as it propagates along the converter is indicated in Fig. 2e.

## Functionalization and characterization of 7,16-dibenzyl-1,4,10,13-tetraoxa-7,16-diazacyclooctadecane (DBTDA) crown ether functionalized chip via Fischer esterification and amine conjugation

The functionalization protocol can be divided into four steps (Fig. 3a) with cleaning of excess reactants implemented after every step. Firstly, a layer of $SiO_2$ (20 nm) is deposited onto silicon via Atomic Layer Deposition (ALD) in the pretreatment step. Secondly, the Fischer esterification method[26] is used to couple the silanol group on the $SiO_2$ surface of the SiP slot waveguides (in the sensing region) with polyacrylic acid $((C_3H_4O_2)_n)$, in the presence of a tin catalyst[53]; specifically tin (II) chloride ($SnCl_2$) dissolved in DI water. The photonic chips, immersed in the reagent, was maintained at 318 K and agitated, while the reaction proceeds for 60 minutes. The chips are then cleaned with DI water. Following, the photonic chips are functionalized with the DBTDA crown ether, forming amide bonds via conjugation, in the

presence of 1-Ethyl-3-(3-dimethylaminopropyl)-carbodiimide) (EDC) and N-Hydroxysuccinimide (NHS) dissolved in ethanol[11,54]. In the final step, the functionalized photonic chips are rinsed in dilute nitric acid ($HNO_3$) to remove as much of tin catalyst adsorbed on the surface as possible. This was done by placing the functionalized photonic chips in a solution of 0.1 M $HNO_3$, agitated with a magnetic stir bar for 60 minutes. Details pertaining to the chemicals used for functionalization is elucidated in the Methods section.

XPS analysis was used to characterize the functionalized photonic chips before functionalization, as well as before and after interaction with different ions to analyze the resulting elemental constitution. A comparison of the N $1s$ region[55] of the XPS spectra before and after functionalization (Fig. 3b) shows that the presence of nitrogen is only detected after the functionalization process. Furthermore, a comparison of the C $1s$ region[55] of the XPS spectra before and after functionalization (Fig. 3c) shows a larger peak area and chemical shift of C $1s$ carbon towards higher binding energies (-286.5 eV[55], -288.8 eV[55]) following functionalization. The binding energy level at 286.5 eV indicates higher carbon concentration and significant carbon binding with electronegative species such as oxygen (C-O) and nitrogen (C-N)[55], and the peak at -288.8 eV corresponds to C = O[55]. Furthermore, the comparison of the O $1s$ region[55] of the XPS spectrums before and after functionalization (Fig. 3d) shows a decrease in oxygen signal and chemical shift of O $1s$ towards a lower binding energy.

This observation can be attributed to the functional layer consisting of more carbon compared to oxygen[55], and the Fischer esterification of silanol with the carboxylic acid group forms O-C bonds, which have a lower binding energy compared to O-Si bonds[55]. The above shows significant evidence that Fischer esterification have been successfully achieved following the elucidated protocol (Fig. 3a). To assess the uniformity of functionalization, in Fig. 3b–d, the N $1s$, C $1s$ and O $1s$ regions respectively of the XPS spectra were taken at four different points (Point 1, 2, 3, 4) on the photonic chip spaced more than 1 cm apart on the photonic chip, showing high consistency. This implies that a functional layer with good uniformity have been realized. In addition, Energy Dispersive X-ray (EDX) analysis pertaining to the functional layer was performed and included in Supplementary Note 6. The results provide compelling evidence for the uniform functionalization of the DBTDA crown ether, which contains carbon bonded to nitrogen and oxygen, via Fischer esterification and amine conjugation.

$Na^+$, $K^+$, $Mg^+$, $Li^+$, $Zn^{2+}$, $Ca^{2+}$, $Fe^{2+}$, $Cu^{2+}$, $Al^{3+}$, $Sn^{2+}$, $Cd^{2+}$, and $Pb^{2+}$ were chosen as highly relevant analytes to quantify the selectivity of the photonic-based detection platform. The selected ions possess a variety of ionic sizes and charge. $Na^+$, $K^+$, $Ca^{2+}$, $Mg^+$, $Zn^{2+}$ and $Cu^{2+}$ are commonly found in bottled water sources[56], while $Fe^{2+}$, $Li^+$, and $Al^{3+}$ could be present in groundwater sources[57]. $Sn^{2+}$ is used as a catalyst in the Fischer esterification process[53], and $Cd^{2+}$ and $Pb^{2+}$ are toxic heavy metals that should be prohibited in drinking water[58]. Each functionalized photonic chip interacts with of the abovementioned ions (DI water, pH = 6.8) independently for 120 s in a microfluidic chamber; see Methods for analyte preparation. After which, the analyte is flushed with DI water and dried with $N_2$ gas blow. XPS is utilized to identify the elemental constitution on the surface of the photonic chips before and after ion interaction via the respective elemental binding energies of each element. Normalization was carried out where the narrow scan XPS spectra prior to ion interaction was subtracted from that after ion interaction. The normalized narrow scan XPS spectra of $Na^+$, $K^+$, $Mg^+$, $Li^+$, $Zn^{2+}$, $Ca^{2+}$, $Fe^{2+}$, $Cu^{2+}$, $Al^{3+}$, $Sn^{2+}$, and $Cd^{2+}$ are displayed in Fig. 4a–k, respectively, indicating the absence of binding on the functionalized photonic sensor. For the abovementioned ions, we note that only $Sn^{2+}$, which is used as the catalyst during Fischer esterification, have been identified prior to ion interaction (see Supplementary Note 7). Conventionally, Fischer esterification is favored when $H_2O$ is removed as the reaction proceeds (dehydrative esterification). However, the

developed esterification process in this work utilizes $H_2O$ as a green solvent, which will decrease the catalytic activity of Brønsted acid catalysts (i.e., $H_2SO_4$)[59]. To that effect (Fig. 3a), the $H_2O$-tolerant Lewis acid catalyst $SnCl_2$, is used[53,60] where Sn is embedded into the $SiO_2$, functioning as a heterogeneous catalyst in the process. It is known that heterogeneous catalysts show improved catalytic activity[61] that favors esterification even in the presence of $H_2O$[62]. This is verified in Fig. 3b–d and Supplementary Note 6, 7. In Fig. 4i, unmistakable binding of $Pb^{2+}$ ions are demonstrated, indicating the presence of $Pb^{2+}$ binding events on the functional layer, via identification of the Pb $4f_{5/2}$ and Pb $4f_{7/2}$ elemental binding energies[55]. Furthermore, in Supplementary Note 6, EDX analysis is performed, where the absence and presence of $Pb^{2+}$ can be clearly seen before and after interaction respectively. From the above, it can be anticipated that the photonic sensor will be selective only towards $Pb^{2+}$, where the ion will bind to the functionalized surface, and be present after analyte flushing. Subsequently, the concentration of exposed $Pb^{2+}$ can be inferred from photonic surface sensing via the shift ($\lambda_s$-$\lambda_0$) in the interferometric spectrum.

## $Pb^{2+}$ photonic sensor characterization

In Fig. 5a, we show the measured fundamental TE mode transmission of the photonic sensor, exposed to DI water (pH = 6.8); see Methods for maintenance of fundamental TE. Details of SiP chip fabrication is elaborated in Supplementary Note 8. As predicted, good interference fringe visibility is obtained, at $\lambda = 1531.9$ nm, where extinction ratio exceeds 20 dB. In contrast, when the sensing region is exposed to air, poor visibility is observed (see Supplementary Note 9). The sensor visibility is reduced at $\lambda = 1563.7$ nm, attributed by the lower water absorption at $\lambda = 1563.7$ nm. The experimental demonstration of the sensor visibility is limited by fabrication bias from design parameters. This would induce changes in designed splitting ratios ($S_1,S_1',S_2,S_2'$), as well as loss difference between the reference and sensing arms. The measured free-spectral range (FSR) is 31.7 nm.

The calibration curve of the $Pb^{2+}$ sensor, indicating ($\lambda_s$-$\lambda_0$) as a function of cumulative $Pb^{2+}$ concentration within the analyte (5, 25, 125, 625, 2625, 12,625, 62,625, 262,625 ppb) is indicated in Fig. 5b through a cumulative testing process (see Methods) with the associated error bars, obtained via six independent measurements ($n = 6$); the inset shows ($\lambda_s$-$\lambda_0$) within a range of 1 to 10 ppb. The pH of the analyte utilized in Fig. 5 is maintained at 6.8 in DI water. A fit was performed between the measured values to understand the form of the calibration curve. The mathematical relationship between ($\lambda_s$-$\lambda_0$) and exposed $Pb^{2+}$ reference concentration is indicated in Supplementary Note 10. The sigmoidal curve is characteristic of (i) absorption isotherms[63,64], attributed by the binding of $Pb^{2+}$ ions to the crown ether functional layer, and (ii) light-matter interaction between the waveguide mode and the functional layer with $Pb^{2+}$ binded. In Fig. 5c, we show a set of MZI spectra around the minima transmission points for the fringes corresponding to each of the abovementioned concentrations. As a large cumulative concentration range is presented in Fig. 5c, the spectra when cumulative $Pb^{2+}$ concentration are 0, 5, and 25 ppb are shown (Fig. 5d). Indicated by the shaded region in Fig. 5b, ($\lambda_s$-$\lambda_0$) exceeds a single FSR when the cumulative concentration of $Pb^{2+}$ is higher than ~4000 ppb. Saturation of ($\lambda_s$-$\lambda_0$) against cumulative concentration is observed in Fig. 5b. This is ascribed to the saturation of the binding sites within the functional layer[63,64]. The lower limit and upper limit of detection are 0.882 ppb and 62600 ppb, respectively, as defined in Supplementary Note 11.

To affirm reproducibility of the calibration curve in Fig. 5b, twenty-four photonic sensors were tested independently, six times each ($n = 6$) at reference concentrations of 1, 10, 80 ppb and 62 ppm. In Fig. 5e, an instance of the transmission spectra when the sensing region is exposed to DI water, DI water containing 80 ppb $Pb^{2+}$, as well as after analyte flush of $Pb^{2+}$ are shown. Based on the positions of $\lambda_s$ in relative to $\lambda_0$, a ($\lambda_s$-$\lambda_0$) of 2.89 nm is obtained. This shift was observed

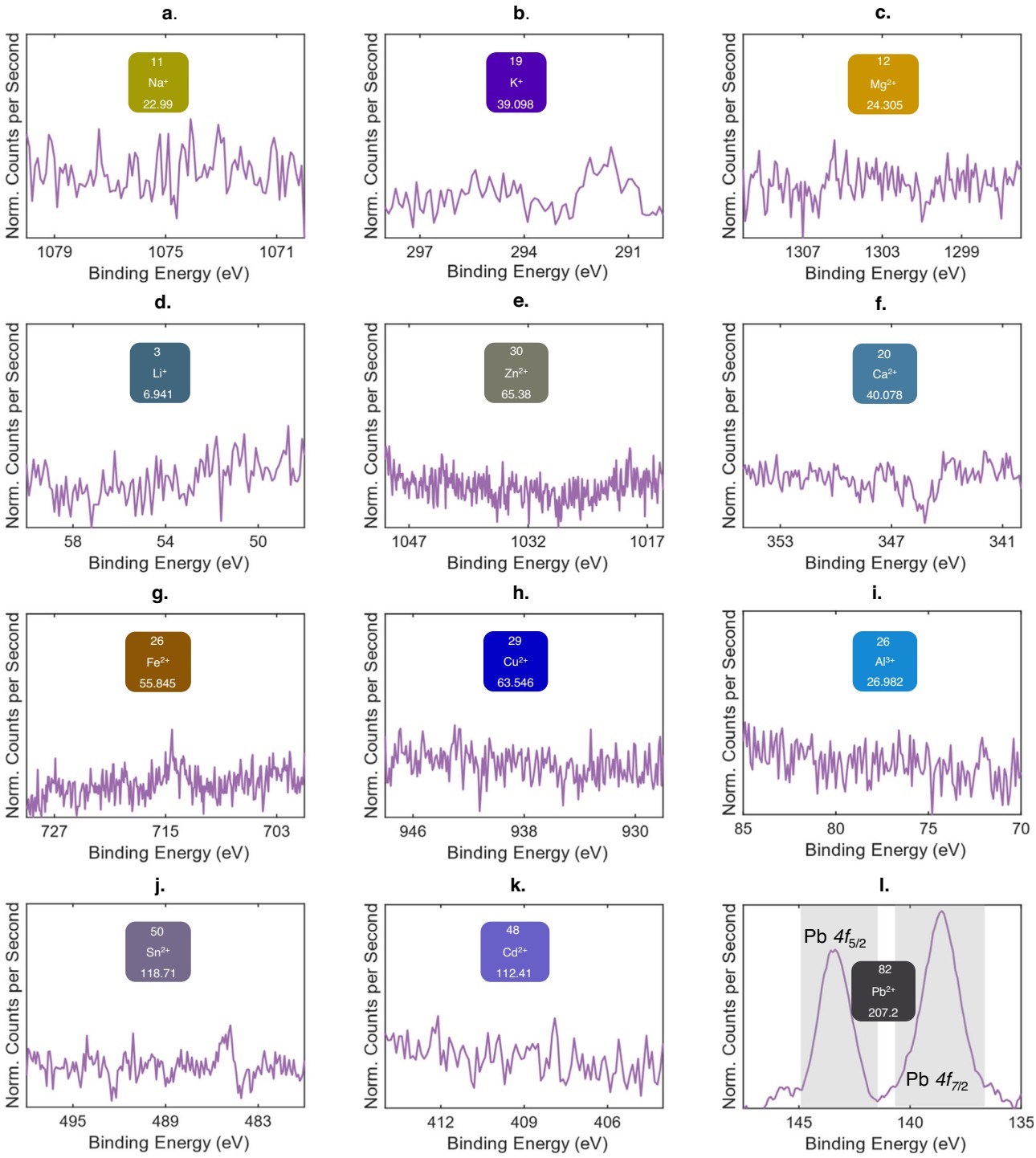

**Fig. 4 | Analysis of the crown ether functional layer selectivity via XPS.** Normalized narrow scan XPS spectra at the photonic chip surface by subtracting the spectra prior to ion interaction from that of after ion interaction (normalization). The ions tested are **a**, Na⁺, **b**, K⁺, **c**, Mg²⁺, **d**, Li⁺, **e**, Zn²⁺, **f**, Ca²⁺, **g**, Fe²⁺, **h**, Cu²⁺, **i**, Al³⁺, **j**, Sn²⁺, **k**, Cd²⁺, and **l**, Pb²⁺ at 100 ppb, in DI water. The pH of all the analyte is maintained at 6.8 (see Methods).

after analyte flushing, consistent with the binding mechanism on the surface of the sensor. By comparing this value to Fig. 5b, the inferred concentration from the calibration curve is 81.7 ppb. Repeated measurements ($n = 6$) presented in Table 1 indicate a mean inferred concentration (Conc._mean) of 79.0 ppb with a standard deviation ($\sigma_{conc.}$) of 1.7 ppb, which is close to the reference value of 80 ppb. The concentration detection error (Err.), defined as the difference in mean sensor inferred concentration from the reference value, is 1.3%. Similarly, Fig. 5f–h shows one of the repeated measurements, performed at

ground truths of 1, 10 ppb and 62 ppm, yielding sensor-inferred concentrations (Fig. 5b) of 1.7, 10.6 ppb and 62.9 ppm, respectively. Likewise, the sensor detection accuracy and repeatability subject to the abovementioned reference standard solutions via multiple measurements ($n = 6$) are presented in Table 1, where Conc._mean/ $\sigma_{conc.}$/ Err. of 1.24 ppb/0.26 ppb/24%, 10.2 ppb/0.3 ppb/ 2% and 64.6 ppm/4.5 ppm/4.2% are inferred respectively. Due to the large detection dynamic range of the sensor, it is imperative to assess the sensor detection accuracies by absolute detection accuracy near the lower

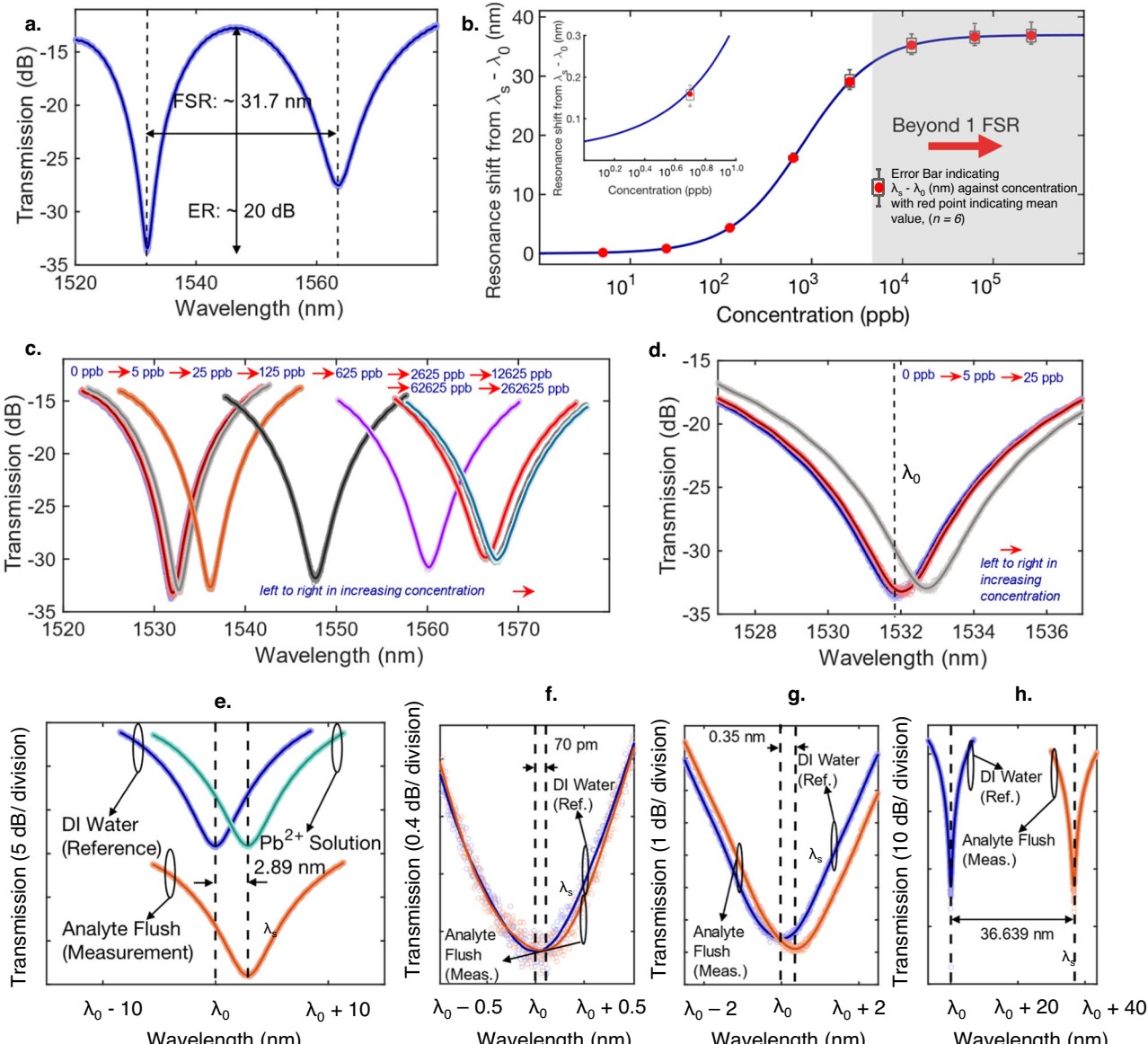

**Fig. 5 | Experimental characterization of Pb²⁺ sensor performance. a** Wavelength spectrum of the sensor, when DI water is applied into the sensor assembly (as shown in Fig. 1d). **b** Calibration curve of the sensor when exposed to reference Pb²⁺ concentration of 0, 5, 25, 125, 625, 2625, 12625, 62625, 262625 ppb via a cumulative testing approach (see Methods). Error bars pertaining to six independent measurements ($n = 6$) are indicated, where the bottom and top edges of the boxplot correspond to the 25th and 75th percentile of the data. The minimum and maximum points of the data are indicated by the extension to which the whiskers of the boxplot are extended to. The mean of the data is indicated by the red point. **c** A set of optical fringe minima corresponding to the tested concentrations in the calibration curve (Fig. 5b). **d** Fig. 5c, zoomed-in at concentrations of 0, 5, 25 ppb. Validation of the calibration curve (Fig. 5b) at reference concentrations of **e**, 80 ppb, **f.** 1 ppb and **g**, 10 ppb, **h**, 62 ppm. For the detection of Pb²⁺ concentration via the sensor in Fig. b–h, the pH of the analyte is maintained at 6.8 (see Methods). DI water is used.

detection limit, and fractional detection limit approaching the upper detection limit. The large detection dynamic range of the sensor (Figs. 5b, e–h, Table 1), implies broad applicability within the Pb²⁺ sensing application space: from monitoring of drinking water (<15 ppb[8]) to heavy industrial applications such as mining[39], smelting[40], battery manufacturing[40] and effluent monitoring[40] where Pb²⁺ concentrations in the tens of ppm is possible. A major benefit of the developed photonic sensors is predicated on its compatibility with scalable and cost-effective processes. This can potentially facilitate the single-use application of the sensors. Alternatively, Supplementary Note 12 also indicates the capacity for reuse.

In Fig. 4 (XPS analysis), the functional layer is found to be selective to Pb²⁺ ions against other tested ions, implying the selectivity of the photonic sensor. To further verify sensor selectivity performance at

the EPA Pb²⁺ safety threshold[8], Na⁺, K⁺, Mg²⁺, Li⁺, Zn²⁺ Ca²⁺, Fe³⁺, Cu²⁺, Al³⁺, Sn²⁺, Cd²⁺ are all tested in DI water at reference concentrations of 15 ppb, where the pH of the analyte is maintained at 6.8. The reference interferometric spectrum (in DI water), and the resulting spectrum after exposure and analyte flush for Cd²⁺ and K⁺ are presented in Fig. 6a, b respectively. The data for the rest of the ions are presented in Supplementary Note 13. No shifts in the interferometric spectra indicative of ion binding are observed. Similar to Fig. 5e–h, six independent photonic sensors were tested at 15 ppb of Pb²⁺. In Fig. 6c, an instance of the measured spectra is shown, where a sensor inferred concentration of 15.0 ppb corresponding to 0.504 nm of ($\lambda_s$-$\lambda_0$) was obtained from Fig. 5b. Repeated measurements ($n = 6$) indicates Conc.ₘₑₐₙ/ $\sigma_{conc}$/ Err. of 14.3/0.6 ppb/4.7% as summarized in Table 1. The results in Fig. 6 and Supplementary Note 13 underpins the ability

of the sensor to effectively detect $Pb^{2+}$ ions in the presence of the other ions.

## Device performance with field samples and speciation of $Pb^{2+}$ ions

The pH tolerance of the $Pb^{2+}$ photonic sensor is of great significance in regard to deployment in practical environmental scenarios. Most environmental water sources where $Pb^{2+}$ are present have a pH range between 6 to 8[42]. It is imperative for the sensor to facilitate accurate and reliable measurements across this pH spectrum. Pb precipitates at higher pH (>9), facilitating the removal of Pb via precipitatation[41]. While the pH of a sample can be adjusted prior to sensor detection, these processing steps are non-trivial for the untrained personnel and hence can limit the scale of adoption with regards to a sensor technology. In the following section, the pH resilience of the $Pb^{2+}$ photonic sensor technology is evaluated. First of all, the capacity of the crown ether functional layer for $Pb^{2+}$ binding across a pH range of 2 − 8 in DI water (pH = 6.8) is investigated via XPS. In Fig. 7a, the normalized $Pb$

$4f_{5/2}$ and Pb $4f_{7/2}$ elemental binding energies are shown. It can be seen that the XPS spectra remains similar within a pH range of 6−8. The introduction of H+ ions within an acidic environment promotes the protonation of oxygen and nitrogen atoms of the crown ether[65], and a reduction in the electron density of the oxygen and nitrogen atoms results. This decreases the availability of coordination to $Pb^{2+}$. Moreover, the addition of positive ions will repel the positively charged $Pb^{2+}$ [66]. The solubility of $Pb^{2+}$ ions decrease significantly at higher pH via precipitation (i.e., $Pb(OH)_2$)[41] where the accurate aqueous detection of $Pb^{2+}$ becomes solubility-limited. The solubility of $Pb^{2+}$ in water sources is beyond the scope of this study. To that effect, the influence of pH on sensor detection accuracy is assessed at environmentally relevant pH values[42] of 5, 6, 6.8, and 8 in DI water, at a reference $Pb^{2+}$ concentration of 15 ppb (EPA limit)[8].

A testing protocol identical to the above (Fig. 1e, Fig. 5) is implemented, where DI water (pH = 6.8) is used to obtain the sensor reference wavelength ($\lambda_0$) and subsequent shift ($\lambda_s$-$\lambda_0$) after analyte flush. Similarly, the sensor inferred concentration is determined from Fig. 5b. Figure 7b shows the measured sensor spectra before ($\lambda_0$) and after analyte flush ($\lambda_s$) at analyte pH values of 5, 6, 6.8, and 8, indicating a ($\lambda_s$-$\lambda_0$)/sensor inferred concentration (Fig. 5b) of 0.278 nm/8.4 ppb, 0.512 nm/15.3 ppb, 0.504 nm/15 ppb and 0.502 nm/15 ppb respectively. In order to further validate the photonic sensor accuracy with changing pH, six separate sensors ($n = 6$) are measured at each pH value, where Conc.mean, $\sigma_{conc.}$ and Err. are indicated in Table 2. As a comparison to Fig. 5, where the pH of the analyte is 6.8, similar levels of detection accuracies are observed at pH of 6−8. However, a noticeable drop in detection accuracy can be seen when the pH is lower than 6 defining the lower pH operating range of the sensor. This is anticipated, and in-line with the XPS measurement at Fig. 7a. The pH operating range (6−8) of the photonic sensor provides coverage for common environmental conditions[42], indicating its versatility to be applied in a multitude of situations without the need for pH control.

To evaluate the effectiveness of the photonic sensor in real-life applications, the technology was deployed to quantify field-collected samples. First of all, tap, lake, and sea water are subjected to ICP-MS for elemental analysis, where the Pb concentrations are found to be 2, 5,

**Table 1 | Detection accuracy and repeatability (Conc.mean/ $\sigma_{conc.}$/Err.) of the crown ether decorated SiP $Pb^{2+}$ sensor against reference $Pb^{2+}$ concentrations, in DI water, verified by ICP-MS**

| Reference $Pb^{2+}$ concentration in analyte (DI water, pH = 6.8) | Inferred Conc.mean/ $\sigma_{conc.}$ ($n = 6$)/ Err. |
|---|---|
| 1 ppb | 1.24 ppb/ 0.26 ppb/24.0% |
| 10 ppb | 10.2 ppb/ 0.3 ppb/2.0% |
| 15 ppb | 14.3 ppb/ 0.6 ppb/4.7% |
| 80 ppb | 79.0 ppb/ 1.7 ppb/1.3% |
| 62 ppm | 64.6 ppm/ 4.5 ppm/4.2% |

The pH of the analyte is 6.8 (see Methods). Conc.mean refers to mean sensor inferred concentrations. $\sigma_{conc.}$ refers to the standard deviation of the sensor-inferred concentrations. Err. refers to the variation in Conc.mean from the reference concentrations. Each reference $Pb^{2+}$ concentration was measured using six independent photonic sensors ($n = 6$).

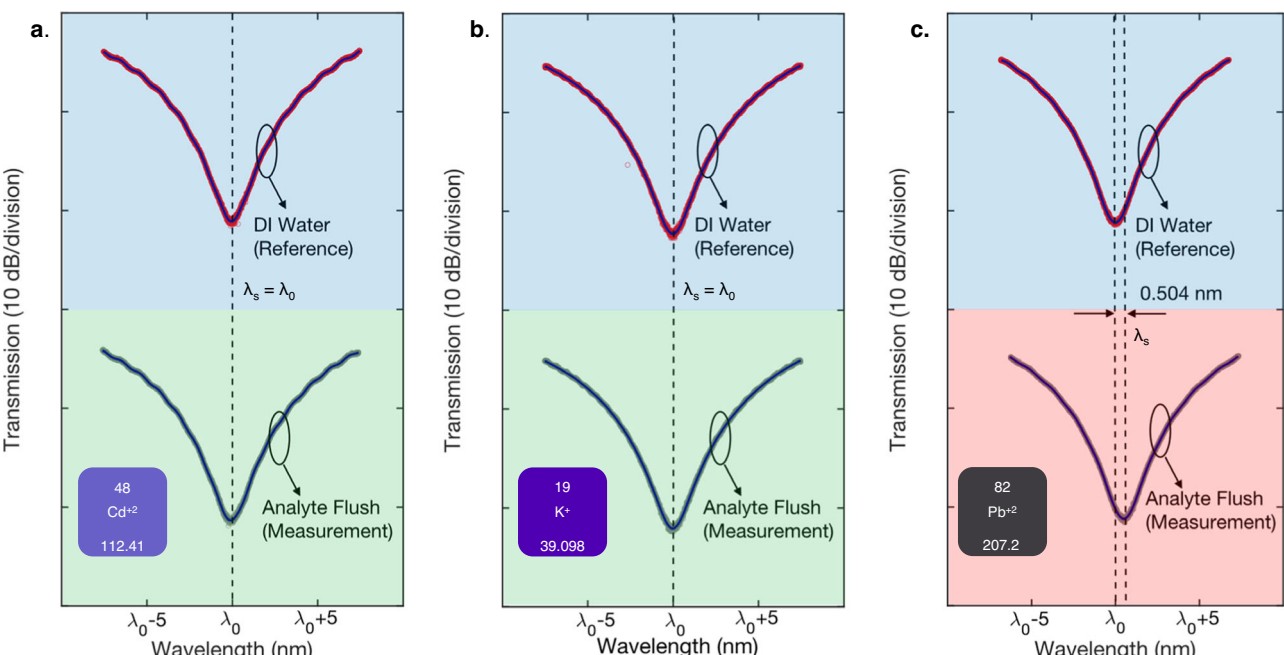

**Fig. 6 | Selectivity performance of the $Pb^{2+}$ ion photonic sensor against. a** $Cd^{2+}$, and **b**, $K^+$ at reference concentrations of 15 ppb in DI water where no shifts in the interferometric spectra indicative of ion binding is observed. **c** The detection performance of the $Pb^{2+}$ photonic ion sensor is evaluated at reference $Pb^{2+}$ concentrations of 15 ppb in DI water. The pH of the analyte is maintained at 6.8 (see Methods).

and 8 ppb, respectively (Supplementary Note 14). The determined pH for tap, lake, and sea water are 7.26, 7.91, and 7.79, correspondingly. Each of the environmental samples are exposed to the sensor for the measurement of $Pb^{2+}$ concentration levels and detection accuracies in the complex matrix. An identical sensor protocol is implemented as above with the exception of the analyte. DI water (pH = 6.8) is used to obtain the sensor reference spectrum ($\lambda_O$), and subsequent spectrum ($\lambda_s$) during analyte flush. The exposed analytes are tap, lake and seawater. $Pb^{2+}$ concentration is inferred from the calibration curve (Fig. 5b) via ($\lambda_s$-$\lambda_O$).

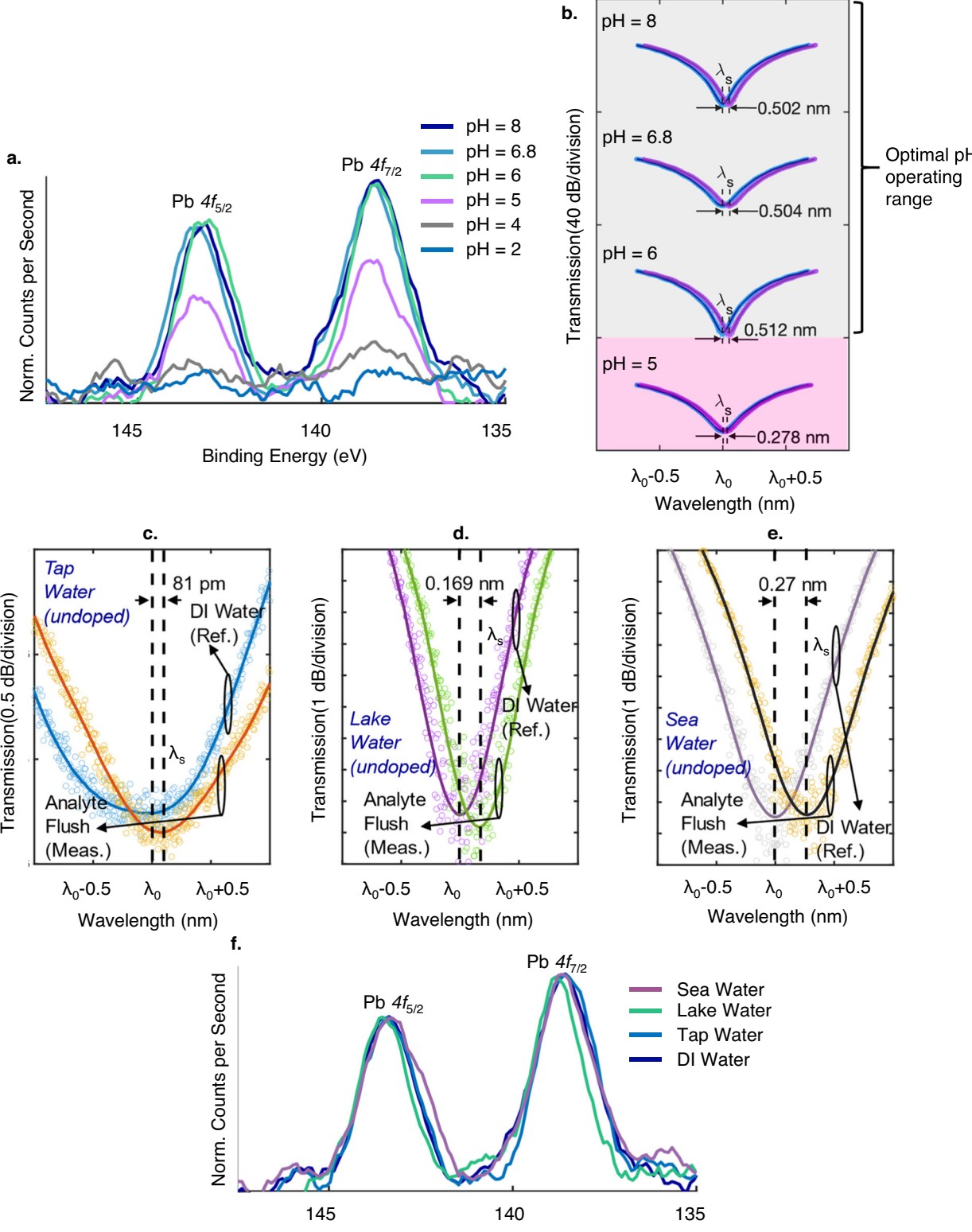

**Fig. 7 | Sensor pH analysis and deployment in field samples. a** $Pb^{2+}$ XPS analysis of the crown ether functional layer across a pH range of 2−8. **b** Optical fringe minima corresponding to 15 ppb of $Pb^{2+}$ reference concentrations in DI water, with pH of 5, 6, 6.8 and 8. Optical fringe minima corresponding to the detection of $Pb^{2+}$ via the photonic sensor in **c**, tap, and **d**, lake, and **e**, sea water. **f** XPS analysis of the crown ether functional layer when the sensor is subjected to DI, tap, lake and seawater.

Figure 7c–e shows the measured $(\lambda_s\text{-}\lambda_0)$ of 0.081 nm for tap water (Fig. 7c), 0.169 nm for lake water (Fig. 7d), and 0.27 nm for sea water (Fig. 7e), corresponding to sensor inferred concentrations (Fig. 5b) of 2.23, 5.06, and 8.16 ppb respectively. For further verification, the concentration of $Pb^{2+}$ in each of these samples were synthetically increased by 15 ppb. Instances of these spectral responses are provided in Supplementary Note 15, showing $(\lambda_s\text{-}\lambda_0)$/ sensor inferred concentrations of 0.570 nm/16.9 ppb, 0.679/20.0 ppb and 0.788 nm/ 23.1 ppb for tap, lake and sea water respectively. To quantify the detection accuracy and repeatability of the sensor in each of the environmental samples, original and synthetically doped, repeated independent measurements ($n = 6$) are implemented using separate sensors, where the Conc.$_{mean}$/ $\sigma_{conc.}$/ Err. are presented in Table 3. The sensor-inferred concentrations matches closely to the ground truth values. A comparison of the detection accuracy to $Pb^{2+}$ ions in DI water (Table 1) indicates similar levels of detection accuracy as well. Further investigation into the crown ether functional layer was conducted where four separate photonic chips were exposed to $Pb^{2+}$ in DI, tap, lake and seawater, and the Pb $4f_{5/2}$ and Pb $4f_{7/2}$ elemental binding energies measured via XPS. In Fig. 7f, the normalized XPS spectrum are found to be similar across all four scenarios. Accordingly, it can be concluded that the $Pb^{2+}$ photonic sensor is resilient to elemental and organic contaminates in view of the strong affinity between the DBTDA crown ethers and $Pb^{2+}$ ions. The pH neutrality of the environmental samples as abovementioned indicates the tendency for $Pb^{2+}$ complexes to exist in larger proportion, in contrast to free ions. The demonstrated ability of the sensor to accurately determine $Pb^{2+}$ concentrations levels even at complexed $Pb^{2+}$ states imply the viability of the sensor in detecting various forms of $Pb^{2+}$. This is attributed to the capacity of the crown ether to displace weaker binding counterions or ligands, occupying the positions where the $Pb^{2+}$ can form coordination bonds with the crown ether[67,68]. This highlights one of the essential advantages of the sensor platform where sample preparation involving multiple steps is not required; for instance, liberation of $Pb^{2+}$ ions from its complexes using acid digestion[69,70].

## Discussion

In this work, crown ether functionalization via Fischer esterification and subsequent amine conjugation is integrated with highly scalable inorganic SiP. This realizes an integrated chip-scale photonic sensing platform that enables the selective binding of $Pb^{2+}$ ions, and subsequent detection down to the ppb-scale. The reaction pathway proposed and demonstrated, driven via Fischer esterification, defies prior expectations that the process is restricted to organics[26]. This enables the engineering of the platform to selectively detect a plethora of ions via subsequent amine conjugation of various crown ethers[27–37]. Furthermore, the functionalization process, by virtue of being solution-based, can be implemented at the wafer-scale. The reactants are dissolved in green solvents, which results in minimal environmental impact. The sensor presented in this work indicates pH resilience (6–8), with the ability to detect $Pb^{2+}$ concentrations in-operando, through a wide dynamic range (1–62,000 ppb), while being highly-selective against other commonly-found, relevant ions. In addition, the viability of the sensor technology in real-life applications is demonstrated through measuring field-collected samples under environmentally relevant conditions. This work represents an encouraging step toward the ubiquitous implementation of photonic-based sensors that protects against widespread $Pb^{2+}$ poisoning. We envisage that this platform can be extended to multiplex ion detection in multiple application spheres.

## Methods

### Microfluidic chamber fabrication

A custom-made acrylic top enclosure, polydimethylsiloxane (PDMS) gasket, and bottom mount make up the flow channel assembly (see Supplementary Note 3). This allows the sample solution and DI water to flow across the sensor on the photonic chips, and doubles as a containment to allow a fixed volume of sample solution to stay atop the sensor for 120 s during ion interaction. The custom-made PDMS gasket is fabricated by curing a PDMS and photo-initiator mixture (Shin-Etsu KER-4690) in polytetrafluoroethylene (PTFE) mold under a 405 nm UV lamp for 10 minutes.

### Chemicals used in photonic chips functionalization

In the Fischer esterification step, polyacrylic acid $((C_3H_4O_2)_n)$ product number 323667 CAS Number: 9003-0, from Sigma-Aldrich, is coupled with the silanol group on the $SiO_2$ surface on the photonic chips, in the presence of tin (II) chloride $(SnCl_2)$ catalyst, product number 323667 CAS Number: product number 208256 CAS Number: 9003-0, with a purity of 98%, from Sigma-Aldrich. The amine conjugation of the selected DBTDA crown ether, product number 294721.

**Table 2 | Detection accuracy and repeatability (Conc.$_{mean}$/ $\sigma_{conc.}$/ Err.) of the crown ether decorated SiP $Pb^{2+}$ sensor against analyte pH, using DI water, with reference $Pb^{2+}$ concentrations of 15 ppb, verified by ICP-MS**

| Analyte pH (DI water, reference $Pb^{2+}$ concentration = 15 ppb) | Inferred Conc.$_{mean}$/ $\sigma_{conc.}$ ($n = 6$)/ Err. |
|---|---|
| pH = 5 | 8.2 ppb/ 0.5 ppb/45.3% |
| pH = 6 | 14.7 ppb/ 0.5 ppb/2.0% |
| pH = 6.8 | 15.5 ppb/ 0.7 ppb/3.3% |
| pH = 8 | 14.8 ppb/ 0.6 ppb/1.3% |

Conc.$_{mean}$ refers to mean sensor inferred concentrations. $\sigma_{conc.}$ refers to standard deviation of the sensor-inferred concentrations. Err. refers to the variation in Conc.$_{mean}$ from the reference concentrations. Each analyte pH was measured using six independent photonic sensors ($n = 6$).

**Table 3 | Detection accuracy and repeatability (Conc.$_{mean}$/ $\sigma_{conc.}$/ Err.) of the crown ether decorated SiP $Pb^{2+}$ sensor in environmental-related scenarios (tap, lake, sea water), using the respective samples, where the $Pb^{2+}$ concentrations are verified by ICP-MS**

| Field samples | Inferred Conc.$_{mean}$/ $\sigma_{conc.}$ ($n = 6$)/ Err. |
|---|---|
| Tap water, undoped; 2 ppb of $Pb^{2+}$, measured by ICP-MS | 2.40 ppb/ 0.3 ppb/20.0% |
| Lake water, undoped; 5 ppb of $Pb^{2+}$, measured by ICP-MS | 4.77 ppb/ 0.6 ppb/4.6% |
| Sea water, undoped; 8 ppb of $Pb^{2+}$; measured by ICP-MS | 8.16 ppb/ 0.4 ppb/2.0% |
| Doped tap water, where $Pb^{2+}$ concentration is synthetically increased by 15 ppb to 17 ppb, measured by ICP-MS | 16.7 ppb/ 0.5 ppb/1.8% |
| Doped lake water, where $Pb^{2+}$ concentration is synthetically increased by 15 ppb to 20 ppb, measured by ICP-MS | 19.5 ppb/ 0.7 ppb/2.5% |
| Doped sea water, where $Pb^{2+}$ concentration is synthetically increased by 15 ppb to 23 ppb, measured by ICP-MS | 22.8 ppb/ 0.7 ppb/0.9% |

Conc.$_{mean}$ refers to mean sensor inferred concentrations. $\sigma_{conc.}$ refers to the standard deviation of the sensor-inferred concentrations. Err. refers to the variation in Conc.$_{mean}$ from the reference concentrations. The measured pH of the tap, lake and sea water are 7.26, 7.91 and 7.79, respectively.

CAS Number: 69703-25-9, with a purity of 97%, from Sigma-Aldrich, with the carboxylic acid group, is carried out in the presence of 1-Ethyl-3-(3-dimethylaminopropyl)-carbodiimide) (EDC) product number 03450, CAS Number: 25952-53-8, with a purity of ≥98%, from Sigma-Aldrich and N-Hydroxysuccinimide (NHS) product number 130672 CAS Number:

6066-82-6, with a purity of 98%, from Sigma-Aldrich dissolved in ethanol, product number 459844 CAS Number: 64-17-5, with a purity of ≥99.5%, from Sigma-Aldrich.

In the final rinsing step, 0.1 M nitric acid ($HNO_3$) used to remove as much of tin catalyst adsorbed on the surface as possible was prepared by diluting concentrated HNO3, product number 438073 CAS Number: 7697-37-2, with a purity of 70%, from Sigma -Aldrich.

### Analyte preparation
For the measurements performed in Figs. 4–6, Supplementary Note 6 and 13, the analyte solution preparation is carried out by diluting 1000 ppm ICP standard solution of the selected ions (TraceCERT® from Merck) with DI water to the concentration required. For low concentrations (lower than 10 ppm), multiple rounds of dilution were performed. ICP-MS is used to verify the concentrations. The pH of the analyte is maintained at 6.8, verified through a pH meter.

To determine the selectivity of the crown ether decorated SiP sensing platform, the functional layer (Fig. 3) and the photonic sensor (Fig. 6) are tested against $Na^+$, $K^+$, $Mg^{2+}$, $Li^+$, $Zn^{2+}$, $Ca^{2+}$, $Fe^{2+}$, $Cu^{2+}$, $Al^{3+}$, $Sn^{2+}$, $Cd^{2+}$ and $Pb^{2+}$ in DI water at concentrations of 100 and 15 ppb respectively. The concentrations are verified using ICP-MS. The dilution methods is elaborated above, in DI water.

To determine sensor detection accuracy as a function of pH (Fig. 7a, b), sodium hydroxide (NaOH), product number 221465 CAS Number: 1310-73-2, with a purity of ≥97%, from Sigma -Aldrich, and nitric acid ($HNO_3$), product number 438073 CAS Number: 7697-37-2, with a purity of 70%, from Sigma -Aldrich, are utilized in DI water to obtain analyte with pH ranging from 2 to 8 first, verified using a pH meter. Following, the dilution methods to 15 ppb of $Pb^{2+}$ remain the same as above. The concentrations are verified using ICP-MS.

For the deployment of the sensor in environmentally related situations (Fig. 7c–f, Supplementary Note 15), water samples are collected from the tap, lake, and sea. The tap water is collected from the washroom in Nanyang Technological University, School of Electrical and Electronic Engineering, 50 Nanyang Avenue, 639798, Singapore. The lake and sea water sources are Jurong Lake Garden, 104 Yuan Ching Road, 618665, Singapore, and West Coast Park Beach, West Coast Ferry Road, 126978, Singapore, respectively. The pH of the tap, lake and sea water are 7.26, 7.91, and 7.79, respectively. ICP-MS is utilized to assess the concentration of the elemental distribution of the samples (See Supplementary Note 14). The concentration of $Pb^{2+}$ in tap, lake and sea water are found to be 2, 5 and 8 ppb, respectively. For further testing of the environmental samples, the concentration of $Pb^{2+}$ are synthetically increased by 15 ppb; implemented via the dilution methods mentioned above. The ground truth $Pb^{2+}$ concentration values of the synthetically doped tap, lake and sea water are 17, 20, and 23 ppb, verified by ICP-MS.

### Fundamental TE maintenance
The $Pb^{2+}$ photonic sensor is designed for fundamental TE operation. Fundamental TE operation is crucial for the maintenance of interference fringes corresponding to the mode, as well as fringe visibility. In order to ensure that the device is operating with only the fundamental TE mode, we utilized a chain of cascaded Multi-Mode Interferometer (MMI) structures that is optimized for the desired polarization (10 × MMI). The polarization-dependent loss the TM mode experience over TE is 2 dB per MMI. Cascading 10 MMIs yields a TM against TE polarization extinction ratio of 20 dB. By optimizing the

input polarization corresponding to the maximum optical power at the output, we will be able to ensure that the device operates with only the fundamental TE mode.

### Measurement of the calibration curve via a cumulative approach
The following is implemented for the measurement of the calibration curve. $Pb^{2+}$ ions are diluted in DI water. The pH of the following analyte is maintained at 6.8. DI water was first added to obtain the reference wavelength ($\lambda_0$), and then flushed from the microfluidic chamber (Step 1). Next, DI water containing $Pb^{2+}$ was added and held for 120 s to facilitate the binding of $Pb^{2+}$ to the functional surface at the sensing region (Step 2). The analyte was then removed again and DI water is added, where the resonant wavelength ($\lambda_s$) is measured to remove the unbound species (Step 3). The shift in wavelength is determined by ($\lambda_s$-$\lambda_0$). For the 6 concentrations that were measured (5, 25, 625, 2625, 62,625, 262,625 ppb), an additive approach was used. In Step 2, DI water containing 5, 20, 600, 2000, 60,000, 200,000 ppb of $Pb^{2+}$ is added sequentially as the exposed concentration is increased. The measurement was repeated six times ($n = 6$) for each set of concentrations involving each photonic sensors, as indicated by the associated error bars in Fig. 5b.

### Reporting summary
Further information on research design is available in the Nature Portfolio Reporting Summary linked to this article.

### Data availability
The data supporting the findings of this study are available from the article and its Supplementary Information. Due to competing interests with regards to the attempts at the commercialization of this sensor technology by Fingate Technologies Pte. Ltd. and Vulcan Photonics SDN. BHD., the source data is available from the corresponding author upon request.

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

## Acknowledgements

Jia Xu Brian Sia would like to acknowledge Ministry of Education/NTU College of Engineering International Postdoctoral Fellowship. This work was performed in part through the use of the facilities of MIT.nano, Harvard University Center for Nanoscale Systems (CNS), NTU's Centre for Micro- & Nano-Electronics (CMNE) and Nanyang Nanofabrication Centre (N2FC). The authors also wish to acknowledge this collaborative work with Fingate Technologies and Vulcan Photonics.

## Author contributions

J.X.B.S., L.R., Y.Z.T. conceived the idea to integrate crown ethers with silicon photonics for ion detection with the help from C.S.O., X.G., K.N.K., X.L., W.W., S.S., C.L., R.R., C.G.L., G.T.R., J.J.H., H.W., Y.Z.T. conceived the functionalization process flow. L.R. did the photonic design and fabrication. X. G. did the photonic sensor characterization. K.N.K. did the material characterization with the help from Y.Z.T. and C.S.O. Y.Z.T. and G.X. designed and constructed the sensor assembly. L.R., Y.Z.T., J.J.H. and J.X.B.S. wrote the manuscript with inputs from C.S.O., X.G., K.N.K., X.L., W.W., S.S., C.L., R.R., C.G.L., G.T.R., H. J.X. B.S., H.W. and J.J.H. supervised the project.

## Competing interests
