## [Peer Review File · Nature Communications]

REVIEWER COMMENTS

Reviewer #1 (Remarks to the Author):

The manuscript by Luigi Ranno et al reported the Crown ether decorated silicon photonics for safeguarding against lead poisoning, which demonstrates a potential application for the pervasive implementation of safeguards against ubiquitous lead poisoning. The manuscript is well organized with sound logic, it should be of great importance to the environmental and analytic chemistry community. However, at present, there are lack of studies on employing this device to detect the Pb²⁺ within practical environment-relevant water samples, and the device's accuracy and reusability in real environment condition is not clear enough. I would suggest the authors further clarify the following concern before it can be accepted for publication.

1. As the resonance shift and transmission do not linearly respond to the Pb²⁺ concentrations rather than other analytic methods, what is the mathematical relationship between the concentration and resonance shift? What is the concentration detection error within this device?
2. The accuracy of the device is of great importance, in the manuscript although the different concentrations of Pb²⁺ ions within different interference ions have been carefully checked, the performance in real environments relevant solution is not shown here. I would suggest the author check their device performance within environmental-related water solutions. Especially the Pb²⁺ contained solution with rich organic contaminates. Please check the device performance with the other well-established method, to compare the accuracy of the established method.
3. The pH tolerance of the surface chemistry within the device is of great importance, the speciation under different pH for lead will be different. I am interested to see how the device detection accuracy responds to the different pH values.
4. The reusability of the device is of great importance as the fabrication cost of this device is not so simple. I would like to suggest the author check and report the device's reusability under different Pb²⁺ concentration solutions. From the XPS data, it seems that there is residue Pb on the device even after water washing. In this regard, how can the device be reused?
5. The physic-chemical Pb²⁺ sensing mechanism is not so clear to me. As shown in Fig 5. Significant signal response could be resolved within the data, while how this response related to the crown ether decorate on silicon photonic from an electron density point of view, is not clear to the readers. Please elaborate on the mechanism carefully from a molecular orbital or electron density point of view.
6. In the present manuscript, it is not clear to me how much solution is required for each test, I suggest the author add the solution pretreatment and requirement conditions of the reported devices.
7. How does different speciation of Pb²⁺ in aqueous solution influence the detection accuracy and limitations?

Reviewer #2 (Remarks to the Author):

The manuscript looks interesting; but missing many important aspects in terms of sensor performance evaluation in terms of lead measurement in water.

The use of Fischer esterification for the functionalization (amine conjugation) of crown ether on silicon was demonstrated.

There are many electrochemical analytical methods including detecting lead using SWASV which can measure relatively low lead concentration (e.g., 1 ppb) in an easy and simple way. Can you compare your sensor with them? Also, address this technology in the introduction.

Several important characterizations and evaluations were unclear as follows:

- Is there a need to measure the lead with that high concentration, 262,000 ppb?
- What is a waveguide surface? Is this Si-substrate?
- Rapid, how fast? Lifetime?
- Is crown ether-based lead detection reproducible? Is immobilized Pb detachable or permanent?
- How many volumes of samples is required? Is water required filtration (filtration mandatory) to prevent any blocking in the sensor platform?
- What is the LOD? It seems that the calibration curves are not linear or have a narrow working range.
- Have you tried with a real sample of aqueous solution for lead detection? Any other interference from other compounds and ions?
- The sensor looks very complicated in fabrication and operation.

Reorder the figure number in order correctly. It is very confusing when reading the manuscript with this complicated figure order.

The use of crown ether amine conjugation against a wide range of relevant ions shows no binding to other ions which gives the selectivity to lead.

Line 41, "public action dwarfs its impact" This is not clear to understand what you want to address.
Line 88, Check the acronym of Environmental and Energy Law Program (EPA). Or is this a program at EPA? Clarify it.

Line 134, what is "SiP"? need more explanation.

Line 153, Does "H2O cladding" mean that the measurement is conducted in aqueous phase?

Reply to the Reviewer's Comments

Dear Reviewers,

The comments by the Reviewers are indicated in red. The author's comments are in blue. A reference list has been added to this letter and they do not correspond to the reference numbers in the main manuscript or the supplementary. The format of the response is formatted as the following:

Reviewer's comments

Author's reply, where direct changes to the manuscript are described in navy blue

Changes to the main text and the supplementary are referred with regards to the page number at the bottom of the main text, and the particular supplementary section respectively

Revised text in the manuscript and the supplementary

Reviewer 1's comments

"The manuscript by Luigi Ranno et al reported the Crown ether decorated silicon photonics for safeguarding against lead poisoning, which demonstrates a potential application for the pervasive implementation of safeguards against ubiquitous lead poisoning. The manuscript is well organized with sound logic, it should be of great importance to the environmental and analytic chemistry community. However, at present, there are lack of studies on employing this device to detect the Pb²⁺ within practical environment-relevant water samples, and the device's accuracy and reusability in real environment condition is not clear enough. I would suggest the authors further clarify the following concern before it can be accepted for publication."

First of all, we would like to thank Reviewer 1 for the time and insightful comments. We appreciate Reviewer 1's recognition of this manuscript: "*The manuscript is well organized with sound logic, it should be of great importance to the environmental and analytic chemistry community.*". This work involves the foremost demonstration of a crown ether decorated integrated silicon photonic Pb²⁺ detection platform driven by Fischer esterification on SiO₂ (20 nm-thick) surfaces, which are deposited on the silicon waveguide surfaces. This paves the way for subsequent uniform amine conjugation of crown ethers¹⁻¹¹. We note that this development also overcomes the existing notion that Fischer esterification is restricted to organic compounds¹². In this demonstration, the 7,16-Dibenzyl-1,4,10,13-tetraoxa-7,16-diazacyclooctadecane (DBTDA) crown ether is utilized, which facilitates lead (Pb²⁺)-selective ion detection, at the part-per-billion scale (i.e., Fig. 5f of the main text). This serves to potentially address the urgent issue of Pb²⁺ toxification in drinking water which accounts for one million deaths annually¹³, amongst many of its other impacts on society¹⁴⁻¹⁸. The sensing platform demonstrates pH resilience (6 - 8), commensurate with typical environmental

conditions¹⁹, and is found to be viable with environmentally-related water sources (i.e., Fig. 7 of the main text). Furthermore, a wide detection dynamic range, where the upper limit of detection is 62000 ppb is experimentally demonstrated (Fig. 5h of the main text). This extends its application space from societal Pb²⁺ poisoning to heavy industries such as mining²⁰, smelting²¹, battery manufacturing²¹, effluent monitoring²¹.

In the following, we will address Reviewer 1's concerns with regards to deployment in environmental-relevant water samples, including accuracy and reusability. We also note that Fingate Technologies Pte Ltd and Vulcan Photonics SDN. BHD. are currently working on the commercialization of the developed sensor platform and the current work is limited in the scope to a conceptual demonstration rather than indication of readiness for commercial deployment.

The current idea with regards to the potential commercialization of this technology is that the devices will not be reused. This is because silicon photonics, by leveraging on industry-established silicon manufacturing, enables high-precision, large-scale, and low-cost manufacturing²². Furthermore, the crown ether functionalization process is solution-based, which lays the ground for wafer scale functionalization, indicating scalability. However, these devices can be reused by the release of the Pb²⁺ ions from the crown ethers through acidification, which we will address in a subsequent comment posed by Reviewer 1.

1. "As the resonance shift and transmission do not linearly respond to the Pb²⁺ concentrations rather than other analytic methods, what is the mathematical relationship between the concentration and resonance shift?"

We thank the Reviewer 1 for the question. Yes indeed, the calibration curve of the sensor (Fig. 5b), which indicates the resonance shift ($\lambda_s - \lambda_0$) as a function of detected Pb²⁺ concentration is not linear. This is attributed to (i) intrinsic absorption isotherm^{23,24} pertaining to the binding of Pb²⁺ ions within the crown ether functional layer on the sensing region of the photonic sensor and (ii) the light-matter interaction between the waveguide mode and the crown ether functional layer with Pb²⁺ binded.

Fig. 5b of the main text indicates the resonance shift ($\lambda_s - \lambda_0$) of the photonic sensor as a function of reference Pb²⁺ tested concentrations in DI water (pH = 6.8). Due to the significant Pb²⁺ detection dynamic range of the sensor, the x-axis, which denotes reference Pb²⁺ concentrations, is implemented in the log-scale.

Based on the experimental data involving the ($\lambda_s - \lambda_0$) of the integrated photonic sensor as a function of tested reference Pb²⁺ concentrations (Fig. 5b of the main text), the mathematical relationship between the reference concentrations (0, 5, 25, 125, 625, 2625, 12625, 62625, 262625 ppb) and the average resonant wavelength shift ($\lambda_s - \lambda_0$) is found to be:

$$\text{Resonance shift } (\lambda_s - \lambda_0)_{\text{average}} = \frac{a}{1 + \exp(-b \times (\text{conc.} - c))}$$

where *conc.* is the reference concentration of Pb^{2+} ions that the photonic sensor is exposed to. *a*, *b*, *c* are fitting parameters. The shape of the calibration curve is characteristic of absorption isotherms^{23,24}. The function is found to accurately model the resonance shift of the photonic sensor as well as the saturation of the binding sites on the crown ether decorated silicon photonic sensor as the reference Pb^{2+} concentration increases.

In view of the above, we have made the necessary revisions in Page 21 of the main text, and the addition of Supplementary Note 10 under Supplementary.

1.1. What is the concentration detection error within this device?"

We thank Reviewer 1 for the very relevant question which motivated us to better characterize the sensors. The concentration detection error is critical in evaluating the performance of the Pb^{2+} sensor. To that, the concentration detection error of the photonic sensor is assessed using 6 separate and independent photonic sensors ($n = 6$) for each tested reference Pb^{2+} concentrations (1, 10, 15, 80 ppb and 62 ppm) in DI water, DI water of different pH ranging from 6 - 8 (environmentally relevant¹⁹), tap, lake and sea water. With each of these analytes, the shift in sensor resonance, $(\lambda_s - \lambda_0)$ are measured, where the Pb^{2+} concentrations are inferred from the sensor calibration curve (Fig. 5b of the main text). We define the Pb^{2+} detection error, Err., as the difference in the mean sensor inferred Pb^{2+} concentration, $\text{Conc.}_{\text{mean}}$ from the reference concentration values, verified via ICP-MS. Furthermore, the repeatability of the sensors is also assessed via the standard deviation, $\sigma_{\text{conc.}}$ of the sensor inferred concentrations.

Via the measurements of 6 separate and independent photonic sensors ($n = 6$) across several reference Pb^{2+} concentrations in DI water, DI water of different pH (6-8), environmentally-relevant water sources (original, and synthetically increased by 15 ppb), the information regards to mean sensor inferred concentration $\text{Conc.}_{\text{mean}}$, sensor detection accuracy Err. ($\text{Conc.}_{\text{mean}}$ from reference concentrations) and standard deviation of sensor inferred concentration $\sigma_{\text{conc.}}$ are presented in Table 1, 2 and 3 of the main text respectively. Textual elaboration about Table 1 are provided in Page 22 and 23-24 of the main text. On the other hand, the textual context of the data in Table 2 and 3 are furnished in Page 25 and 28-29 of the main text respectively. The tables mentioned are reproduced below:

Reference Pb²⁺ concentration in analyte (DI water, pH = 6.8)	Inferred Conc._{mean}/ $\sigma_{\text{conc.}}$ ($n = 6$)/ Err.
1 ppb	1.24 ppb/ 0.26 ppb/ 24.0 %
10 ppb	10.2 ppb/ 0.3 ppb/ 2.0 %
15 ppb	14.3 ppb/ 0.6 ppb/ 4.7 %
80 ppb	79.0 ppb/ 1.7 ppb/ 1.3 %
62 ppm	64.6 ppm/ 4.5 ppm/ 4.2 %

Table 1. Detection accuracy and repeatability (Conc._{mean}/ $\sigma_{\text{conc.}}$ /Err.) of the crown ether decorated SiP Pb²⁺ sensor against reference Pb²⁺ concentrations, in DI water, verified by ICP-MS. The pH of the analyte is 6.8 (see Methods). Conc._{mean} refers to mean sensor inferred concentrations. $\sigma_{\text{conc.}}$ refers to standard deviation of the sensor inferred concentrations. Err. refers to the variation in Conc._{mean} from the reference concentrations. Each reference Pb²⁺ concentration was measured using six independent photonic sensors ($n = 6$).

Analyte pH (DI water, reference Pb²⁺ concentration = 15 ppb)	Inferred Conc._{mean}/ $\sigma_{\text{conc.}}$ ($n = 6$)/ Err.
pH = 5	8.2 ppb/ 0.5 ppb/ 45.3 %
pH = 6	14.7 ppb/ 0.5 ppb/ 2.0 %
pH = 6.8	15.5 ppb/ 0.7 ppb/ 3.3 %
pH = 8	14.8 ppb/ 0.6 ppb/ 1.3 %

Table 2. Detection accuracy and repeatability (Conc._{mean}/ $\sigma_{\text{conc.}}$ /Err.) of the crown ether decorated SiP Pb²⁺ sensor against analyte pH, using DI water, with reference Pb²⁺ concentrations of 15 ppb, verified by ICP-MS. Conc._{mean} refers to mean sensor inferred concentrations. $\sigma_{\text{conc.}}$ refers to standard deviation of the sensor inferred concentrations. Err. refers to the variation in Conc._{mean} from the reference concentrations. Each analyte pH was measured using six independent photonic sensors ($n = 6$).

Environmental-Related Scenarios	Inferred Conc._{mean}/ $\sigma_{\text{conc.}}$ ($n = 6$)/ Err.
Tap water, undoped; 2 ppb of Pb ²⁺ , measured by ICP-MS	2.40 ppb/ 0.3 ppb/20.0 %
Lake water, undoped; 5 ppb of Pb ²⁺ , measured by ICP-MS	4.77 ppb/ 0.6 ppb/ 4.6 %
Sea water, undoped; 8 ppb of Pb ²⁺ ; measured by ICP-MS	8.16 ppb/ 0.4 ppb/ 2.0 %
Doped tap water, where Pb ²⁺ concentration is synthetically increased by 15 ppb to 17 ppb, measured by ICP-MS	16.7 ppb/ 0.5 ppb/ 1.8 %
Doped lake water, where Pb ²⁺ concentration is synthetically increased by 15 ppb to 20 ppb, measured by ICP-MS	19.5 ppb/ 0.7 ppb/ 2.5 %
Doped sea water, where Pb ²⁺ concentration is synthetically increased by 15 ppb to 23 ppb, measured by ICP-MS	22.8 ppb/ 0.7 ppb/ 0.9 %

Table 3. Detection accuracy and repeatability (Conc._{mean}/ $\sigma_{\text{conc.}}$ / Err.) of the crown ether decorated SiP Pb²⁺ sensor in environmental-related scenarios (tap, lake, sea water), using the respective samples, where the Pb²⁺ concentrations are verified by ICP-MS. Conc._{mean} refers to mean sensor inferred concentrations. $\sigma_{\text{conc.}}$ refers to standard deviation of the sensor inferred concentrations. Err. refers to the variation in Conc._{mean} from the reference concentrations. The measured pH of the tap, lake and sea water are 7.26, 7.91 and 7.79 respectively.

2. *"The accuracy of the device is of great importance, in the manuscript although the different concentrations of Pb²⁺ ions within different interference ions have been carefully checked, the performance in real environments relevant solution is not shown here. I would suggest the author check their device performance within environmental-related water solutions. Especially the Pb²⁺ contained solution with rich organic contaminates.*

We thank Reviewer 1 for the comment. Indeed, the Reviewer is right to point out the importance of assessing the Pb²⁺ detection performance in environmentally-related water solutions, rich in organic contaminates. In order to address the comment, we tested the sensor platform in three different environmental water sources, namely, Singaporean tap, lake and sea water. Organic contaminates are found to exist in environmentally relevant water sources such as Singapore's lake²⁵ and sea water²⁶. The pH of the tap, lake and sea water are found to be 7.26, 7.91 and 7.79 respectively. Minute traces of the by-product of diphenylguanidine (DPG)²⁷ are also found to be present in Singapore's tap water. First of all, the elemental composition of the tap, lake and sea water are assessed via ICP-MS (Supplementary Note 14 of the Supplementary), and found to be 2, 5 and 8 ppb respectively. Following, the photonic sensor reference wavelength (λ_0) is first measured via the exposure of the sensor to DI water. Following, the DI water is flushed out, three independent photonic sensors are exposed to tap, lake and sea water separately for 120 s. After which, the above analyte is flushed out from the microfluidic chamber with DI water and the photonic sensor resonant redshift is measured again (λ_s). The ($\lambda_s - \lambda_0$) are found to be 0.081 nm for tap water (Fig. 7c of the main text), 0.169 nm for lake water (Fig. 7d of the main text), and 0.27 nm for sea water (Fig. 7e of the main text), corresponding to sensor inferred concentrations of 2.23, 5.06, and 8.16 ppb respectively via Fig. 5b of the main text. This indicates a good match between the sensor inferred Pb²⁺ concentration and the references values. In addition, the concentration of Pb²⁺ in tap, lake and sea water are synthetically increased by 15 ppb to 17, 20 and 23 ppb, where a single measurement involving one photonic sensor for each sample is also performed. ($\lambda_s - \lambda_0$)/ sensor inferred concentration of 0.57 nm/16.9 (Supplementary Fig. 11a of the Supplementary), 0.679 nm/20.0 ppb (Supplementary Fig. 11b of the Supplementary) and 0.788 nm/ 23.1 ppb (Supplementary Fig. 11c of the Supplementary) are obtained from the synthetically-doped tap, lake and sea water samples respectively. To investigate the photonic sensor Pb²⁺ detection accuracy and repeatability, each tap, lake and sea water sample, original and synthetically-doped are subject to 6 separate photonic sensors ($n = 6$). The mean sensor inferred concentration, Conc._{mean}, standard deviation of sensor inferred concentrations, $\sigma_{conc.}$, and concentration detection error (Conc._{mean} from reference concentration), Err. are presented in Table 3 of the main text. It is imperative to note that near the lower detection limit of the sensor, absolute sensor detection accuracy is essential. The pH neutrality of the tap, lake and sea water implies the tendency for Pb²⁺ complexes to exist in higher proportion. The ability of the crown ether decorated silicon photonic Pb²⁺ detection platform to detect Pb²⁺, even in the complexed state, implies the agnostic nature of the technology in detecting various forms of Pb²⁺. This

highlights one of the key strengths of the platform where laborious sample preparation is not required.

Mention of validating the crown ether decorated silicon photonic Pb^{2+} sensing platform in environmentally-relevant situations (tap, lake and sea water) is provided in Page 6 and 30 of the main text.

Data with regards to subjecting the Pb^{2+} photonic sensor to tap, lake and sea water are provided in Fig. 7c, d, e of the main text. Data where the Pb^{2+} photonic sensor is subject to synthetically-doped (Pb^{2+} concentrations increased by 15 ppb) are indicated in Supplementary Fig. 11a, b, c of the Supplementary. Via the measurement of 6 separate and independent photonic sensors ($n = 6$) across each of the abovementioned environmentally-relevant water sources (original and synthetically-doped tap, lake and sea water sources), the information regards to mean sensor inferred concentration $\text{Conc.}_{\text{mean}}$, standard deviation of sensor inferred concentration $\sigma_{\text{conc.}}$ and sensor detection accuracy ($\text{Conc.}_{\text{mean}}$ from reference concentration) Err. are presented in Table 3 of the main text. The textual addition are provided in Page 27-29 of the main text. The table and figures mentioned are reproduced below:

Optical fringe minima corresponding to the detection of Pb^{2+} via the photonic sensor in **b**, tap, and **c**, lake, and **d**, sea water. **e**, XPS analysis of the crown ether functional layer when the sensor is subjected to DI, tap, lake and sea water.

Supplementary Fig. 11 Pb²⁺ sensor deployment in environmentally related situations where the concentration of tap, lake and sea water are synthetically increased by 15 ppb to a, 17, b, 20, c, 23 ppb.

The pH of the synthetically doped tap, lake and sea water are 8.26, 7.91, and 7.79 respectively.

Environmental-Related Scenarios	Inferred Conc._{mean}/ $\sigma_{\text{conc.}}$ ($n = 6$)/ Err.
Tap water, undoped; 2 ppb of Pb ²⁺ , measured by ICP-MS	2.40 ppb/ 0.3 ppb/ 20.0 %
Lake water, undoped; 5 ppb of Pb ²⁺ , measured by ICP-MS	4.77 ppb/ 0.6 ppb/ 4.6 %
Sea water, undoped; 8 ppb of Pb ²⁺ ; measured by ICP-MS	8.16 ppb/ 0.4 ppb/ 2.0 %
Doped tap water, where Pb ²⁺ concentration is synthetically increased by 15 ppb to 17 ppb, measured by ICP-MS	16.7 ppb/ 0.5 ppb/ 1.8 %
Doped lake water, where Pb ²⁺ concentration is synthetically increased by 15 ppb to 20 ppb, measured by ICP-MS	19.5 ppb/ 0.7 ppb/ 2.5 %
Doped sea water, where Pb ²⁺ concentration is synthetically increased by 15 ppb to 23 ppb, measured by ICP-MS	22.8 ppb/ 0.7 ppb/ 0.9 %

Table 3. Detection accuracy and repeatability (Conc._{mean}/ $\sigma_{\text{conc.}}$ / Err.) of the crown ether decorated SiP Pb²⁺ sensor in environmental-related scenarios (tap, lake, sea water), using the respective samples, where the Pb²⁺ concentrations are verified by ICP-MS. Conc._{mean} refers to mean sensor inferred concentrations. $\sigma_{\text{conc.}}$ refers to standard deviation. The measured pH of the tap, lake and sea water are 7.26, 7.91 and 7.79 respectively.

Information with regards to the procurement of the environmentally-relevant water sources (tap, lake and sea water) are provided in Page 32 of the main text (see Methods).

2.1. Please check the device performance with the other well-established method, to compare the accuracy of the established method."

We thank Reviewer 1 for the comment. We agree that a comparison of the detection accuracy of the crown ether decorated silicon photonic Pb^{2+} sensing platform with other well-established methods in the Pb^{2+} detection space is essential to provide a well-rounded manuscript. Furthermore, we have added on several more factors that are imperative to a Pb^{2+} detection platform, given the widespread and significant impact that it has on society¹³ (Supplementary Table 1).

In Supplementary Table 1, we provide a comparison of the sensor performance demonstrated in this work, with several well-established technologies. Thus far, in the Pb^{2+} detection space, besides the ICP-MS²⁸ /OES²⁹, there has been significant development in fluorescence^{30–33}, SWASV electrochemical^{34–37}, colorimetric (quantitative)^{38,39}, fiber-based sensors⁴⁰ indicating remarkable performances (Supplementary Table 1). Pb^{2+} detection at the ppb-scale have been achieved with demonstrated viability in environmentally-relevant scenarios. However, it is imperative to note that these technologies represent discrete sensors where analytics are required on a separate platform. It is important to note the availability of such compact analytical instruments. For instance, handheld potentiometers⁴¹ for SWASV electrochemical sensors, and micro-photospectrometer⁴² for fluorescence sensors. However, the significant costs of these instruments can serve to limit the widespread adoption of these Pb^{2+} sensor technologies, specifically in developing nations, where the impact of Pb^{2+} toxification is the most significant^{43,44}. The implementation of optics, for Pb^{2+} detection represents an emerging technology, where the implementation of a fiber interrogator as an analytical instrument can lower the costs required for widespread implementation of Pb^{2+} sensor technologies⁴⁰. However, by harnessing the scalability of manufacturing and integrability of silicon CMOS processing^{45–47}, advancements towards the development of a low-cost, integrated Pb^{2+} sensor technology that is widely implemented can be achieved. A well-known example that highlights the advantages of such integrated silicon systems can be seen from the miniaturization of transistors, which has led to the ubiquitous availability of personal computing. In addition, most of the Pb^{2+} sensor technologies have indicated a fixed, or limited pH range (Supplementary Table 1), which implies requirements on sample processing. Via the first development of the crown ether decorated silicon photonic Pb^{2+} sensing platform, this work enables the realization of Pb^{2+} detection within an integrated silicon platform. A large Pb^{2+} detection dynamic range (1 - 62000 ppb) is demonstrated, with good pH resilience. The photonic platform is also shown to be viable in environmentally-relevant scenarios (tap, lake, sea water). The sensor pH operating range is commensurate with that of typical environmental-related scenarios¹⁹, and its Pb^{2+} detection dynamic range indicates the capacity for it to be used in a multitude of applications: from the monitoring of drinking water to heavy industry (i.e., mining²⁰, smelting²¹, battery manufacturing²¹, effluent monitoring²¹). The integrated Pb^{2+} photonic sensing platform can be integrated with waveguide-based analytical components (i.e.,

spectrometer⁴⁸) on the same silicon photonic platform. Furthermore, by leveraging on the economies of scale that can be derived from silicon manufacturing⁴⁹⁻⁵², these integrated sensor systems can be realized at low costs, without compromises in performance⁴⁵⁻⁴⁷. The abovementioned characteristics of the crown ether decorated silicon photonic Pb²⁺ sensing platform underscores the much-needed proliferation of low-cost, high performance Pb²⁺ chip-scale sensors to safeguard against widespread Pb²⁺ toxification in society.

In Page 4 of the main text, we have pointed to Supplementary Note 1 which seeks to provide an overview and discussion of this developed work and other well-established technologies.

We have added a discussion on the overview of this work, and well-established Pb²⁺ detection technologies in Page 2-3 of the Supplementary under Supplementary Note 1. An overview of the abovementioned technologies is provided in Supplementary Table 1.

3. The pH tolerance of the surface chemistry within the device is of great importance, the speciation under different pH for lead will be different. I am interested to see how the device detection accuracy responds to the different pH values.

We thank Reviewer 1 for the comment. We agree that the pH tolerance of the surface chemistry is of importance to the developed integrated sensor technology, especially when no pre-processing steps are being taken.

This manuscript pertains to the development of a crown ether decorated silicon photonic platform for the aqueous detection of Pb²⁺, which potentially safeguards against widespread societal Pb²⁺ poisoning. First of all, it is of note that environmental water sources have a pH range that ranges between 6-8¹⁹. The solubility of lead is reduced at higher pH where aqueous detection will be lead solubility-limited⁵³ and not limited by the developed sensing platform; the solubility of lead at higher pH is not within the scope of this work. As such, this defines the upper pH range that the Pb²⁺ photonic sensor platform will be tested to (pH = 8). It is of note that the performance of the crown ether decorated silicon photonic platform will be impacted at lower pH due to the introduction of H⁺ ions within an acidic environment that promotes the protonation of oxygen and nitrogen at the 7,16-Dibenzyl-1,4,10,13-tetraoxa-7,16-diazacyclooctadecane (DBTDA) crown ether⁵⁴. This leads to a reduction in the electron density of the oxygen and nitrogen atoms, decreasing the availability of coordination to Pb²⁺ ions. Furthermore, the addition of positive H⁺ ions will also repel the positively charged Pb²⁺.

To that effect, the DBTDA crown ether binding capacity to Pb²⁺ within a pH range of 2-8 is investigated through X-ray Photoelectron Spectroscopy (XPS). In Fig. 7a of the main text, the normalized Pb 4f_{5/2} and Pb4f_{7/2} elemental binding energies are indicated. It can be observed that the XPS spectrum remains similar across a pH range of 6-8. However, at pH of below 5 to

2, the signal strength of the Pb $4f_{5/2}$ and Pb $4f_{7/2}$ elemental binding energies decreases as the pH drops. As abovementioned, this can be attributed to the protonation and reduction in electron density of the nitrogen and oxygen atoms within the DBTDA crown ether through the addition of H^+ , which reduces coordination to Pb^{2+54} . In addition, the H^+ atoms repels the positively charged Pb^{2+} ions which leads to a reduction in the binding capacity of the crown ether to Pb^{2+} .

In view Fig. 7a of the main text, the Pb^{2+} photonic sensors are tested at a pH of 5, 6, 6.8 and 8 in DI water, at Pb^{2+} concentrations of 15 ppb⁵⁵ (EPA limit). In Fig. 7b, an instance of each spectral shifts are indicated. A $(\lambda_s - \lambda_0)$ /sensor inferred concentration of 0.278 nm/8.4 ppb, 0.512 nm/15.3 ppb, 0.504 nm/15.0 ppb and 0.502 nm/15.0 ppb are obtained; the sensor inferred concentrations are obtained from Fig. 5b of the main text through the measured $(\lambda_s - \lambda_0)$. To further validate the observed trend, six independent Pb^{2+} photonic sensors ($n = 6$) are tested at each of the pH value, where the value of mean sensor inferred concentrations, $Conc_{\text{mean}}$, standard deviation of sensor inferred concentration, $\sigma_{\text{conc.}}$, and sensor detection accuracy ($Conc_{\text{mean}}$ from reference concentration), $Err.$ are provided in Table 3 of the main text. A close correspondence to reference Pb^{2+} concentrations (verified via ICP-MS) is obtained at pHs of 8, 6.8, 6 before degrading as the pH decreases to 5.

Mention of the crown ether decorated silicon photonic Pb^{2+} sensing platform pH operating range is highlighted in Page 6, 30 of the main text.

Pb^{2+} XPS analysis of the crown ether functional layer at pH of 2, 4, 5, 6, 6.8 and 8 are added to Fig. 7a of the main text. An instance of optical fringe minima corresponding to 15 ppb of reference Pb^{2+} concentrations in DI water at each pH value of 5, 6, 6.8 and 8 are added to Fig. 7b of the main text. Via the measurement of 6 separate and independent photonic sensors ($n = 6$) across pH of 5, 6, 6.8 and 8, information regards to mean sensor inferred concentration, $Conc_{\text{mean}}$, sensor inferred concentration standard deviation, $\sigma_{\text{conc.}}$ and sensor detection accuracy ($Conc_{\text{mean}}$ from reference concentrations), $Err.$ are presented in Table 2 of the main text. The tables and figures mentioned are reproduced below:

Fig. 7. Sensor pH analysis and deployment in field samples. a, Pb²⁺ XPS analysis of the crown ether functional layer across a pH range of 2 - 8. **b,** Optical fringe minima corresponding to 15 ppb of Pb²⁺ reference concentrations in DI water, with pH of 5, 6, 6.8 and 8.

Analyte pH (DI water, reference Pb ²⁺ concentration = 15 ppb)	Inferred Conc. _{mean} / σ_{conc} . ($n = 6$)/ Err.
pH = 5	8.2 ppb/ 0.5 ppb/ 45.3 %
pH = 6	14.7 ppb/ 0.5 ppb/ 2.0 %
pH = 6.8	15.5 ppb/ 0.7 ppb/ 3.3 %
pH = 8	14.8 ppb/ 0.6 ppb/ 1.3 %

Table 2. Detection accuracy and repeatability (Conc._{mean}/ σ_{conc} ./ Err.) of the crown ether decorated SiP Pb²⁺ sensor against analyte pH, using DI water, with reference Pb²⁺ concentrations of 15 ppb, verified by ICP-MS. Conc._{mean} refers to mean sensor inferred concentrations. σ_{conc} refers to standard deviation. Each analyte pH was measured using six independent photonic sensors ($n = 6$).

Textual information pertaining to the investigation of the crown ether decorated silicon photonic Pb²⁺ sensing platform pH dependence is added in Page 24-25 of the main text.

The preparation of DI water, doped with 15 ppb of Pb²⁺, at pH of 2, 4, 5, 6, 6.8 and 8 are provided in Page 32 of the main text.

4. From the XPS data, it seems that there is residue Pb on the device even after water washing. In this regard, how can the device be reused?"

Yes, Reviewer 1 is right to point out that the Pb^{2+} will remain on the crown ether-based functional layer after analyte flush. This is intentional, as the Pb^{2+} selective DBTDA crown ether will bind to Pb^{2+} ions only during analyte exposure (120s). In the analyte, particles, ions or molecules might be present, which will affect photonic sensor response through evanescent sensing. This will lead to erroneous detection if not removed. The purpose of the analyte flush is to isolate the subsequent photonic response solely to the crown ether-bound Pb^{2+} ions. Detailed information relating to sensor operation is elucidated in Page 7 - 10 of the main text. In our following response to this comment, the Pb^{2+} ions can be released from the crown ethers by subjecting the sensors to 0.1 M of HNO_3 , for 60 minutes. This enables the sensor to be reused (Supplementary Note 12).

4.1. "The reusability of the device is of great importance as the fabrication cost of this device is not so simple. I would like to suggest the author check and report the device's reusability under different Pb^{2+} concentration solutions

We appreciate the Reviewer 1's comment as reusability and general cost are important metrics for the evaluation of an environmental sensor. It is of note that the development of the functionalization process and photonic design is complex. However, the fabrication of the photonic devices leverages on silicon manufacturing, which enables the high precision, large scale manufacturing of these devices at the wafer-scale⁴⁵⁻⁴⁷. Moreover, the crown ether functionalization process is solution-based where reactants are dissolved in green solvents such as water and ethanol. This implies that low-cost wafer-scale functionalization can be readily implemented. The abovementioned implies that the crown ether decorated silicon photonic Pb^{2+} detection platform can be manufactured at scale with low costs.

As mentioned in a previous paragraph replying to your earlier comment, we are currently working with Fingate Technologies Pte Ltd and Vulcan Photonics SDN. BHD. to commercialize the crown ether decorated silicon photonic Pb^{2+} detection platform. Thanks to the scalability, and the low-cost that is inherent to the manufacturing of these devices, our plan is to implement the devices for one-time usage. However, it should be noted that these crown ether decorated silicon integrated Pb^{2+} sensors can be reused.

Specifically, the binding mechanism between Pb^{2+} and the DBTDA crown ethers is facilitated through coordinate covalent bonds that are formed between the electron rich oxygen and nitrogen atoms on the crown ether and the electron deficient Pb^{2+} . This leads to the formation of a stable complex in solution^{56,57}. It is of note that the binding process is optimal when the crown ethers are not protonated. With the addition of H^+ ions into the aqueous environment via

acidification, the protonation of the nitrogen and oxygen atoms on the crown ether occurs⁵⁴. This results in a reduction in the availability of active sites that form coordination bonds with Pb^{2+} . In addition, the introduction of positive charges (H^+) would also lead to the repelling of positively-charged Pb^{2+} . The above leads to a reduction in the binding affinity between Pb^{2+} and the crown ethers. To that effect, Pb^{2+} which are binded to the crown ethers will be released as free ions in solution.

Via the concept above, the reusability of the crown ether decorated silicon photonic Pb^{2+} detection platform is investigated. The reusability is assessed in three cases: 1.) 15 ppb Pb^{2+} reference concentration in DI water, pH = 6.8 (Fig. 6c of the main text), 2.) 80 ppb Pb^{2+} reference concentration in DI water, pH = 6.8 (Fig. 5e of the main text), 3.) 23 ppb Pb^{2+} reference concentration in sea water, pH = 7.79 (Supplementary Fig. 11c of the Supplementary) for two Pb^{2+} Detection Cycles. The data from Case 1 and 3 are replicated in Supplementary Fig. 9a and b respectively under the Detection Cycle 1. On the other hand, the data from Case 2 is replicated in Supplementary Fig. 9c under Detection Cycle 1. The reference resonant wavelength of the first detection cycle and the subsequent measured resonant wavelength upon 120s of Pb^{2+} exposure and analyte flush (defined in Page 7-10 of the main text) are termed as $\lambda_{0,1}$ and $\lambda_{s,1}$. Likewise, the two resonant photonic resonant wavelengths are defined as $\lambda_{0,2}$ and $\lambda_{s,2}$ for the second Detection Cycle. Upon the first detection cycle, where the sensor inferred concentrations (Fig. 5b of the main text) are obtained via $(\lambda_{s,1}-\lambda_{0,1})$, these devices are subjected to 0.1 M of dilute nitric acid (HNO_3), for 60 minutes to regenerate the sensors. The detailed steps are elucidated as follow, starting from the first detection cycle: 1.) The photonic sensors are first exposed to DI water to obtain $(\lambda_{0,1})$, as shown in Supplementary Fig. 9a-c. 2.) The analytes highlighted in case 1, 2, and 3 are exposed to separate photonic sensors, for 120 s. 3.) Analyte flush is performed, where the $\lambda_{s,1}$ is obtained. The sensor inferred concentrations can then be obtained from Fig. 5b of the main text via $(\lambda_{s,1}-\lambda_{0,1})$, where in Supplementary Fig. 9a-c, we show an instance of the spectral shift for the three cases highlighted above. 4.) Following, the devices in the three cases are subjected to 0.1 M of dilute nitric acid (HNO_3), for 60 minutes. The Pb^{2+} ions are released from the crown ethers as free ions and flushed away via DI water. Steps 1-3 are repeated again for Detection Cycle 2. It can be seen that $\lambda_{0,1} \approx \lambda_{0,2}$ in Supplementary Fig. 9a-c, indicating that the sensors have been regenerated through the acidification process where the Pb^{2+} ions are released from the crown ethers leading to a drop in the material index of the crown ether-based functional layer; resonant wavelength blueshift will result. Furthermore, $(\lambda_{s,2}-\lambda_{0,2}) \approx (\lambda_{s,1}-\lambda_{0,1})$, implying a close match in the sensor inferred concentrations between the two detection cycles. For further validation of the protocol, the reusability of the sensors are subjected to 6 Pb^{2+} photonic sensors ($n = 6$), for each of the three analyte cases. Supplementary Table 2 highlights the sensor inferred concentrations, $\text{Conc.}_{\text{mean}}$, sensor inferred concentration standard deviation, $\sigma_{\text{conc.}}$, and concentration detection error, Err. , where a close match between the $\text{Conc.}_{\text{mean}}$ of the first, second detection cycle and the reference Pb^{2+} concentrations (verified via ICP-MS) is obtained. The reusability of the devices are validated across different Pb^{2+} reference concentrations.

An instance Pb^{2+} photonic sensor resonant wavelength shift ($\lambda_s - \lambda_0$), indicating reusability in 1.) 15 ppb reference Pb^{2+} concentration in DI water (pH = 6.8), 2.) 80 ppb reference Pb^{2+} concentration in DI water (pH = 6.8), and 3.) 23 ppb reference Pb^{2+} concentration in sea water (pH = 7.79) have been included in Supplementary Fig. 9. The mentioned figures are reproduced below:

Supplementary Fig. 9 Demonstration of sensor reusability. The optical fringe minima measured in DI water to determine $\lambda_{0,1}$, $\lambda_{0,2}$ and after analyte flush to determine $\lambda_{s,1}$, $\lambda_{s,2}$ performed over two detection cycles at Pb^{2+}

concentration of **a**, 15 ppb in DI water (pH = 6.8) **b**, 23 ppb in sea water (pH = 7.79) **c**, 80 ppb in DI water (pH = 6.8). The optical fringe minima for $\lambda_{0,1}$ and $\lambda_{s,1}$ in a-c corresponds to the spectrum measured at reference and after analyte flush in Fig. 6c of the main text, Supplementary Fig. 11c, and 5e of the main text respectively. The optical fringe minima for $\lambda_{0,2}$ is measured after sensor regeneration through exposure to 0.1 M of HNO₃ for 60 minutes. The optical fringe minima for $\lambda_{s,2}$ is measured after 120 s of Pb²⁺ exposure for each of the abovementioned tested analyte and followed by analyte flush.

Following the abovementioned protocol for sensor reuse, 6 independent photonic sensors are used to validate the process under each of the three analyte cases. Supplementary Table 2 have been added, under Supplementary Note 12, which indicates the sensor mean sensor inferred concentrations, Conc._{mean}, sensor inferred concentration standard deviation, σ_{conc} , and sensor detection accuracy Err. (Conc._{mean} from reference concentrations) between two Pb²⁺ detection cycles. The table mentioned is reproduced below:

Reference concentration and analyte pH	Pb ²⁺ = δ) from the first detection	Inferred Conc. _{mean} / σ_{conc} . (n = δ) from the first detection	Inferred Conc. _{mean} / σ_{conc} . (n = δ) from the second detection
15 ppb in DI water, pH = 6.8		15.5 ppb/ 0.7 ppb/ 3.3 %	15.4 ppb/ 0.6 ppb/ 2.7 %
23 ppb in synthetically doped sea water; concentration of Pb ²⁺ increased by 15 ppb. pH = 7.79		22.8 ppb/ 0.7 ppb/ 0.9 %	22.6 ppb/ 0.8 ppb/ 1.7 %
80 ppb in DI water, pH = 6.8		80.9 ppb/ 2.0 ppb/ 1.1 %	82.4 ppb/ 2.9 ppb/ 3.0 %

Supplementary Table 2 Demonstration of sensor detection accuracy and reusability. Pb²⁺ concentrations are 15 ppb in DI water (pH = 6.8), 23 ppb in sea water (pH = 7.79) and 80 ppb in DI water (pH = 6.8) over two detection cycles. Conc._{mean} refers to mean sensor inferred concentrations. σ_{conc} refers to standard deviation of the sensor inferred concentrations. Err. refers to the variation in Conc._{mean} from the reference concentrations. To investigate the repeatability of the protocol, six separate sensors ($n = 6$) are tested at each concentration. The reference Pb²⁺ concentrations are verified via ICP-MS.

Contextual information with regards to Supplementary Fig. 9 and Supplementary Table 2 have been added to Page 22-23 of the main text, and Supplementary Note 12.

5. *"The physic-chemical Pb²⁺ sensing mechanism is not so clear to me. As shown in Fig 5. Significant signal response could be resolved within the data, while how this response related to the crown ether decorate on silicon photonic from an electron density point of view, is not clear to the readers. Please elaborate on the mechanism carefully from a molecular orbital or electron density point of view."*

We thank Reviewer 1 for the comment. This would enable the readers to better understand the detection mechanism of the crown ether decorated silicon photonic Pb²⁺ detection platform.

To provide more clarity with regards to the operating mechanism of the developed sensor, the following changes have been made:

- Illustration indicating the binding, and subsequent detection of Pb²⁺ ions via the crown ethers have been improved to more closely resemble reality (Fig. 1a of the main text)
- An additional zoomed-in image of the sensing region have been added, where the technology sub-layers of the crown ether decorated silicon photonic Pb²⁺ detection platform are highlighted: silicon photonics (SiP), Fischer esterification, and crown ether functional layer (Fig. 1b of the main text)

In the following, we will elucidate on the Pb²⁺ sensing mechanism from an electron density point of view. The sensor operating protocol is illustrated in Fig. 1e of the main text. First of all, the Pb²⁺ sensor is exposed to DI water. This facilitates the extraction of the reference wavelength (λ_0), where all subsequent sensor resonant shift (λ_s) will be referenced to it. Following, the DI water is flushed, and analyte possibly containing Pb²⁺ ions will be added into the microfluidic chamber, and exposed for 120 s. Should the analyte contain Pb²⁺ ions, the Pb²⁺ ions will bind to the crown ethers and be immobilised on the crown ether-based functional layer within the sensing region of the photonic sensor. The Pb²⁺ complexation with its ligand can be explained through the interaction of the 6s² electrons and the accepting 6p orbitals with the electron lone pairs on the donating oxygen and nitrogen groups on the crown of the crown ether^{58,59}. Following, the analyte will be flushed from the microfluidic chamber. This is to isolate the subsequent optical response of the photonic sensor to Pb²⁺ ions, through the removal of other particles/ions/molecules that may be present in the analyte solution. More simply, the flushing step will extract all the dissolved or suspended contaminants in the analyte solution with the exception of the Pb²⁺ which are bound to the crown ethers. DI water is added, and the subsequent resonant wavelength will be measured (λ_s). When Pb²⁺ ions binds to the crown ether-based functional layer (Fig. 1b of the main text), the high electron density of Pb²⁺ increases the polarizability of the functional layer. This results in a rise in the relative permittivity of the functional layer, which in turn increases the effective and group index of the waveguide mode. As the group index of the sensing arm is larger than that of the reference arm,

an increase in the material index of the crown ether-based functional layer, implemented within the sensing will lead to the redshift of λ_s , relative to λ_0 . Subsequently, the sensor inferred concentration can be obtained via $(\lambda_s - \lambda_0)$ from the calibration curve in Fig. 5b of the main text. The operating mechanism of the sensor platform as elucidated above, is demonstrated experimentally within this work.

Amendments have been made to the illustration of the crown ether decorated silicon photonic Pb^{2+} detection platform (Fig. 1a of the main text) to better enable the understanding of the Pb^{2+} sensor mechanism. Furthermore, a zoomed-in illustration of the sensing region (Fig. 1b of the main text) has been added together with the technology sub-layers of the platform. This also serves to enable the readers to better understand the operating mechanism of the photonic platform.

Illustration pertaining to the operating mechanism of the crown ether decorated silicon photonic Pb^{2+} detection platform is provided in Fig. 1a, b, and e of the main text. The mentioned figures are reproduced below:

Fig. 1 Concept of the Crown Ether/SiP platform for Pb^{2+} ion detection. **a**, 3-D illustration of the photonic Pb^{2+} ion sensor based on the crown ether decorated SiP platform. The functionalization performed in the sensing region is indicated. For sake of illustration, the 20 nm SiO_2 deposited on top of the waveguides in the sensing region is not indicated. Information is provided in Fig. 3a. **b**, The zoomed-in illustration of the sensing region, where the binding between the Pb^{2+} ions and crown ether functional layer is depicted. The technology sub-layers in the sensing region are indicated: SiP, Fischer esterification, crown ether functional layer. **c**, The micrograph image of the Pb^{2+} ion sensor, where the sensing arm and scale bar (500 μm) are indicated. **d**, The Pb^{2+} photonic sensor assembly, consisting of the photonic chip and a microfluidic chamber. **e**, Elucidated operating principle of the

photonic Pb^{2+} ion sensor. The inset shows the exemplary applications that the ion detection platform can be extended to.

Contextual information relating to the careful description of the sensor Pb^{2+} detection mechanism is added in Page 7 - 10 of the main text.

6. *"In the present manuscript, it is not clear to me how much solution is required for each test, I suggest the author add the solution pretreatment and requirement conditions of the reported devices."*

We thank Reviewer 1 for the comment. The microfluidic channel is designed to hold 0.426 ml of solution. No pretreatment of the analyte is required, with the exception of filtering through a syringe filter (0.45 μm). The sensor has been tested extensively against various ions (Fig. 6a, b of the main text, Supplementary Fig. 10a-i) where it is found to be selective only to Pb^{2+} (Fig. 6c of the main text). Furthermore, the sensor is also found to be viable in environmentally-relevant water sources (tap, lake and sea water) in Fig. 7c-f of the main text, Supplementary Fig. 11a-c, and Table 3 of the main text.

The sensor has a pH operating range of 6 - 8 where no pH treatment is required (Fig. 7a-b of the main text). The upper pH operating range is limited by the solubility of the Pb and not the developed technology itself. As such, the accurate aqueous detection of Pb^{2+} ions in solution will be Pb solubility-limited⁵³ and not by the developed sensing platform. On the other hand, the lower pH operating range is limited by the crown ethers, where the reason is furnished in our response to *Comment 3*.

Mention that sample pretreatment is not required is furnished in Page 6 and 29 of the main text.

We have added that the microfluidic channel holds 0.426 ml of solution in Page 10 of the main text.

We have added that a syringe filter (0.45 μm) is used to filter the analyte in Page 10 of the main text.

We have mentioned that the sensor have a pH range of 6 - 8 (limited by the solubility of Pb at higher pH) in Page 6, 24, 25, 30 of the main text, Page 3 of the Supplementary and Supplementary Table 1. The sensor do not require pH treatment is required within a pH range of 6 - 8.

7. "How does different speciation of Pb^{2+} in aqueous solution influence the detection accuracy and limitations?"

We thank Reviewer 1 for the comment. It is pertinent to consider how the different speciation of Pb^{2+} in aqueous solution will affect the detection accuracy, and what is the limitations of the crown ether decorated silicon photonic Pb^{2+} detection platform.

To that effect, we have considered how pH impacts the performance of the Pb^{2+} photonic detection platform. This is elucidated in our reply in *Comment 3*.

Furthermore, the viability of the Pb^{2+} photonic detection platform is also considered in environmentally-related water sources (tap, lake and sea water), where organic contaminants are present. This has been addressed in *Comment 2* of our reply.

Mention of the crown ether decorated silicon photonic Pb^{2+} sensing platform pH operating range is highlighted in Page 6, 30 of the main text.

Pb^{2+} XPS analysis of the crown ether functional layer at pH of 2, 4, 5, 6, 6.8 and 8 are added to Fig. 7a of the main text. An instance of optical fringe minima corresponding to 15 ppb of reference Pb^{2+} concentrations in DI water at each pH value of 5, 6, 6.8 and 8 are added to Fig. 7b of the main text. Via the measurement of 6 separate and independent photonic sensors ($n = 6$) across pH of 5, 6, 6.8 and 8, information regards to sensor inferred concentration, $Conc_{mean}$, sensor inferred concentration standard deviation, σ_{conc} , and sensor detection accuracy Err . ($Conc_{mean}$ from reference concentrations) are presented in Table 2 of the main text.

Textual information pertaining to the investigation of the crown ether decorated silicon photonic Pb^{2+} sensing platform pH operating range is added in Page 24-25 of the main text.

The preparation of DI water, doped with 15 ppb of Pb^{2+} , at pH of 2, 4, 5, 6, 6.8 and 8 are provided in Page 32 of the main text.

Mention of validating the crown ether decorated silicon photonic Pb^{2+} sensing platform in environmentally-relevant situations (tap, lake and sea water) is provided in Page 2, 6 and 30 of the main text.

Data with regards to subjecting the Pb^{2+} photonic sensor to tap, lake and sea water are provided in Fig. 7c, d, e, f of the main text. Data where the Pb^{2+} photonic sensor is subject to synthetically-doped (Pb^{2+} concentrations increased by 15 ppb) are indicated in Supplementary Fig. 11a, b, c of the Supplementary. Via the measurement of 6 separate

and independent photonic sensors ($n = 6$) across each of the abovementioned environmentally-relevant water sources (original and synthetically-doped tap, lake and sea water sources), the information regards to mean sensor inferred concentrations $\text{Conc.}_{\text{mean}}$, standard deviation of sensor inferred concentrations $\sigma_{\text{conc.}}$ and sensor detection accuracy $\text{Err.}(\text{Conc.}_{\text{mean}}$ from reference concentrations) are presented in Table 3 of the main text. The textual addition are provided in Page 27-29 of the main text. All of the abovementioned tables and figures are reproduced below:

Fig. 7. Sensor pH analysis and deployment in field samples. **a.** Pb²⁺ XPS analysis of the crown ether functional layer across a pH range of 2 - 8. **b.** Optical fringe minima corresponding to 15 ppb of Pb²⁺ reference concentrations in DI water, with pH of 5, 6, 6.8 and 8. Optical fringe minima corresponding to the detection of Pb²⁺ via the

photonic sensor in **b**, tap, and **c**, lake, and **d**, sea water. **e**, XPS analysis of the crown ether functional layer when the sensor is subjected to DI, tap, lake and sea water.

Analyte pH (DI water, reference Pb^{2+} concentration = 15 ppb)	Inferred $\text{Conc.}_{\text{mean}} / \sigma_{\text{conc.}} (n = 6) / \text{Err.}$
pH = 5	8.2 ppb/ 0.5 ppb/ 45.3 %
pH = 6	14.7 ppb/ 0.5 ppb/ 2.0 %
pH = 6.8	15.5 ppb/ 0.7 ppb/ 3.3 %
pH = 8	14.8 ppb/ 0.6 ppb/ 1.3 %

Table 2. Detection accuracy and repeatability ($\text{Conc.}_{\text{mean}} / \sigma_{\text{conc.}} / \text{Err.}$) of the crown ether decorated SiP Pb^{2+} sensor against analyte pH, using DI water, with reference Pb^{2+} concentrations of 15 ppb, verified by ICP-MS. $\text{Conc.}_{\text{mean}}$ refers to mean sensor inferred concentrations. $\sigma_{\text{conc.}}$ refers to standard deviation of the sensor inferred concentrations. Err. refers to the variation in $\text{Conc.}_{\text{mean}}$ from the reference concentrations. Each analyte pH was measured using six independent photonic sensors ($n = 6$).

Environmental-Related Scenarios	Inferred $\text{Conc.}_{\text{mean}} / \sigma_{\text{conc.}} (n = 6) / \text{Err.}$
Tap water, undoped; 2 ppb of Pb^{2+} , measured by ICP-MS	2.40 ppb/ 0.3 ppb/ 20.0 %
Lake water, undoped; 5 ppb of Pb^{2+} , measured by ICP-MS	4.77 ppb/ 0.6 ppb/ 4.6 %
Sea water, undoped; 8 ppb of Pb^{2+} ; measured by ICP-MS	8.16 ppb/ 0.4 ppb/ 2.0 %
Doped tap water, where Pb^{2+} concentration is synthetically increased by 15 ppb to 17 ppb, measured by ICP-MS	16.7 ppb/ 0.5 ppb/ 1.8 %

Doped lake water, where Pb^{2+} concentration is synthetically increased by 15 ppb to 20 ppb, measured by ICP-MS	19.5 ppb/ 0.7 ppb/ 2.5 %
Doped sea water, where Pb^{2+} concentration is synthetically increased by 15 ppb to 23 ppb, measured by ICP-MS	22.8 ppb/ 0.7 ppb/ 0.9 %

Table 3. Detection accuracy and repeatability ($Conc_{mean}/\sigma_{conc}/Err.$) of the crown ether decorated SiP Pb^{2+} sensor in environmental-related scenarios (tap, lake, sea water), using the respective samples, where the Pb^{2+} concentrations are verified by ICP-MS. $Conc_{mean}$ refers to mean sensor inferred concentrations. σ_{conc} refers to standard deviation of the sensor inferred concentrations. $Err.$ refers to the variation in $Conc_{mean}$ from the reference concentrations. The measured pH of the tap, lake and sea water are 7.26, 7.91 and 7.79 respectively.

Information with regards to the procurement of the environmentally-relevant water sources (tap, lake and sea water) are provided in Page 32-33 of the main text (see Methods).

**Reviewer 2's comment**

The manuscript looks interesting; but missing many important aspects in terms of sensor performance evaluation in terms of lead measurement in water.

We thank Reviewer 2 for the supportive comment " *The manuscript looks interesting* ". In the following, we will address each one of your comments in detail.

The use of Fischer esterification for the functionalization (amine conjugation) of crown ether on silicon was demonstrated

We thank Reviewer 2 for the recognition of this innovation. We note that a 20 nm SiO₂ layer is deposited via Atomic Layer Deposition (ALD), on the silicon slot waveguides within the sensing region. This process is illustrated in Fig. 3a of the main text. Through the process flow developed in Fig. 3a of the main text, this work overcomes the conventional view that Fischer esterification is restricted to organics only. From a broader perspective, the successful Fischer of an inorganic material (SiO₂) implies the agnostic nature of this process, potentially displacing silylation agents with trisubstituted silyl groups, which are typically used to couple silica/silicon with organic compounds. It is pertinent to note that these reagents can potentially self-reaction, leading to agglomeration, and hence decreasing the uniformity of functionalization, negatively impacting sensor reproducibility.

With the successful uniform demonstration of Fischer esterification on SiO₂ (Fig. 3b-d of the main text), different crown ethers, selective to various ions (i.e., K⁴, Be⁵, Ra⁷, Cs²), as illustrated at the inset of Fig. 1e, can undergo amine conjugation following Fischer esterification on SiO₂. This indicates far reaching applications (i.e., medical⁴, electronics manufacturing⁵, nuclear^{2,7}) that the platform can be extended to.

There are many electrochemical analytical methods including detecting lead using SWASV which can measure relatively low lead concentration (e.g., 1 ppb) in an easy and simple way. Can you compare your sensor with them? Also, address this technology in the introduction.

We thank Reviewer 2 for the comment; we certainly agree that a detailed comparison of the developed technology with other methods can improve the quality of the manuscript by enabling the reader to better contextualize the result within the state-of-art. A good overview of existing technologies with this work needs to be furnished for the manuscript to be complete. This will enable readers to understand the advantages and disadvantage of each technology. In the Pb²⁺ detection space, besides ICP-MS/OES, significant advancements have been made in Pb²⁺ sensors based on fluorescence, colorimetry (quantitative), and fiber-based technologies, which have indicated impressive performances. In Supplementary Table 1, we provide an overview of the developed work with the abovementioned technologies. Pb²⁺ detection at the

ppb-scale have been achieved with demonstrated viability in environmentally-relevant scenarios. However, it is imperative to note that these technologies represent discrete sensors where analytics are required on a separate platform. It is important to note the availability of such compact analytical instruments. For instance, handheld potentiometers⁴¹ for SWASV electrochemical sensors, and micro-photospectrometer⁴² for fluorescence sensors. However, the significant costs of these instruments can serve to limit the widespread adoption of these Pb²⁺ sensor technologies, specifically in developing nations, where the impact of Pb²⁺ toxification is the most significant^{43,44}. The implementation of optics, for Pb²⁺ detection represents an emerging technology, where the implementation of a fiber interrogator as an analytical instrument can lower the costs required for widespread implementation of Pb²⁺ sensor technologies⁴⁰. However, by harnessing the manufacture scalability and integrability of silicon manufacturing⁴⁵⁻⁴⁷, advancements towards the development of a low-cost, integrated Pb²⁺ sensor technology that is widely implemented can be achieved. A well-known example that highlights the advantages of such integrated silicon systems can be seen from the miniaturization of transistors, which has led to the ubiquitous availability of personal computing. In addition, most of the Pb²⁺ sensor technologies have indicated a fixed, or limited pH range (Supplementary Table 1), which implies requirements on sample processing. Via the first development of the crown ether decorated silicon photonic Pb²⁺ sensing platform, this work enables the realization of Pb²⁺ detection within an integrated silicon platform. A large Pb²⁺ detection dynamic range (1 - 62000 ppb) is demonstrated, with good pH resilience. The photonic platform is also shown to be viable in environmentally-relevant scenarios (tap, lake, sea water). The sensor pH operating range is commensurate with that of typical environmental-related scenarios¹⁹, and its Pb²⁺ detection dynamic range indicates the capacity for it to be used in a multitude of applications: from the monitoring of drinking water to heavy industry (i.e., mining²⁰, smelting²¹, battery manufacturing²¹, effluent monitoring²¹). The integrated Pb²⁺ photonic sensing platform different from discrete Pb²⁺ detection technologies, can be integrated with waveguide-based analytical components (i.e., spectrometer⁴⁸) on the same silicon photonic platform. Furthermore, by leveraging on the economies of scale that can be derived from silicon manufacturing⁴⁹⁻⁵², these integrated sensor systems can be realized at low costs, without compromises in performance⁴⁵⁻⁴⁷. The abovementioned characteristics of the crown ether decorated silicon photonic Pb²⁺ sensing platform underscores the much-needed proliferation of low-cost, high performance Pb²⁺ chip-scale sensors to safeguard against widespread Pb²⁺ toxification in society.

We have addressed SWASV electrochemical sensors together with fluorescence, colorimetry (quantitative), and fiber-based Pb²⁺ sensors in the introduction of the manuscript (Page 4 of the main text) to facilitate an overview of the existing technologies that are being developed.

We have added a discussion on the overview of well-established Pb²⁺ detection technologies in Page 2-3 of the Supplementary under Supplementary Note 1. An overview of the technologies is provided in Supplementary Table 1.

1. Is there a need to measure the lead with that high concentration, 262,000 ppb?

We thank Reviewer 2 for the comment. We would like to highlight that we have made a typo in defining the upper bound of the sensor detection range. The value should be 62000 ppb instead of 262000 ppb. This unintended mistake was made in the abstract and conclusion section and not in the design and experimental section. The concentration range (5, 25, 126, 625, 2625, 12625, 62625, 262625 ppb) used to obtain Fig. 5b of the main text enables us to understand the form of the calibration curve. The sigmoidal shape of Fig. 5b in the main text is characteristic of absorption isotherms^{23,24}, which is attributed by (i) the binding of Pb²⁺ ions to the crown ether functional layer, and (ii) light-matter interaction between the waveguide mode and the functional layer with Pb²⁺ binded.

For further clarification, we have also added Supplementary Note 11 in the Supplementary, which elucidates about how the sensor Pb²⁺ detection dynamic range is defined (lower and upper limit of detection).

The Pb²⁺ detection dynamic range of the integrated photonic sensor is defined as follows. The standard deviation of the sensor noise (σ_n) is found to be 0.014 nm. This is obtained from 10 repeated resonant wavelength measurements of the photonic sensor, when exposed to DI water. It is important to note that this accounts for intrinsic sensor resonance drift due to environmental thermal fluctuations, and the accuracy of the experimental setup in determining the resonant wavelength. Due to the large Pb²⁺ detection dynamic range of the sensor, it is pertinent to note that the lower and upper detection limits should be assessed by absolute accuracy and fractional accuracy respectively. To that effect, the lower limit of detection is found to be 0.882 ppb, corresponding to $(\lambda_s - \lambda_0)$ of 0.042 nm, which is three times that of the standard deviation of the sensor noise ($3\sigma_n$)³⁷. On the other hand, the upper limit of detection is defined by the assessing the difference in $(\lambda_s - \lambda_0)$ about $\pm 10\%$ of the average resonant wavelength shift $(\lambda_s - \lambda_0)_{\text{average}}$ of the highest tested concentration in Fig. 5b of the main text, that is still larger than $3\sigma_n$ ⁶⁰. This criteria will facilitate the confident quantification of Pb²⁺ concentration within the range. The upper limit of detection is found to be 62000 ppb. The difference in $(\lambda_s - \lambda_0) \pm 10\%$ about $(\lambda_s - \lambda_0)_{\text{mean}}$ is 0.068 nm; the value of $3\sigma_n$ is 0.042 nm. To further validate the abovementioned conditions in defining the Pb²⁺ photonic sensor lower and upper detection limits. Pb²⁺ detection near the lower limits have been performed and validated as shown in Fig. 5f and Table 1 of the main text (1 ppb). The upper Pb²⁺ detection limit is affirmed in Fig. 5h and Table 1 of the main text (62000 ppb) where good fractional detection accuracies to the ground truth concentration values have been obtained.

While the detection of Pb^{2+} concentrations below 15 ppb is pertinent to prevent societal Pb poisoning, having a large Pb^{2+} detection dynamic range enables the extension of the crown ether decorated silicon photonic Pb^{2+} detection platform to multiple Pb^{2+} sensing application spheres. The large upper bound of detection broadens the applicability of the platform to heavy industries such as mining²⁰, smelting²¹, battery manufacturing²¹, effluent monitoring²¹, where detection of Pb^{2+} concentrations at the tens of part-per-millions is mandated.

The photonic sensor Pb^{2+} upper detection limit has been corrected in Page 30 of the main text.

The upper limit of detection (62000 ppb) pertaining to the Pb^{2+} integrated photonic sensor have been validated and added in Fig. 5h and Table 1 of the main text. Textual information is furnished in Page 22 of the main text. The mentioned figures and tables are reproduced below:

Validation of the calibration curve (Fig. 5b) at reference concentrations of **h**, 62 ppm. For the detection of Pb^{2+} concentration via the sensor in Fig. 5b-h, the pH of the analyte is maintained at 6.8 (see Methods). DI water is used.

Reference Pb ²⁺ concentration in analyte (DI water, pH = 6.8)	Inferred Conc. _{mean} / $\sigma_{\text{conc.}}$ ($n = 6$)/ Err.
1 ppb	1.24 ppb/ 0.26 ppb/ 24.0 %
10 ppb	10.2 ppb/ 0.3 ppb/ 2.0 %
15 ppb	14.3 ppb/ 0.6 ppb/ 4.7 %
80 ppb	79.0 ppb/ 1.7 ppb/ 1.3 %
62 ppm	64.6 ppm/ 4.5 ppm/ 4.2 %

Table 1. Detection accuracy and repeatability (Conc._{mean}/ $\sigma_{\text{conc.}}$) of the crown ether decorated SiP Pb²⁺ sensor against reference Pb²⁺ concentrations, in DI water, verified by ICP-MS. The pH of the analyte is 6.8 (see Methods). Conc._{mean} refers to mean sensor inferred concentrations. $\sigma_{\text{conc.}}$ refers to standard deviation of the sensor inferred concentrations. Err. refers to the variation in Conc._{mean} from the reference concentrations. Each reference Pb²⁺ concentration was measured using six independent photonic sensors ($n = 6$).

The definition of the lower and upper bounds of Pb²⁺ detection pertaining to the Pb²⁺ integrated photonic sensor has been added via Supplementary Note 11 of the Supplementary. In the main text, text has been added in Page 21, that points readers to Supplementary Note 11.

Heavy industrial applications of the crown ether decorated silicon photonic Pb²⁺ detection platform has been added in Page 6, 22 of the main text.

2. *What is a waveguide surface? Is this Si-substrate?*

We thank Reviewer 2 for the confirmation. The waveguide surface is SiO₂. We note that 20 nm of SiO₂ have been deposited via Atomic Layer Deposition (ALD) in Fig. 3a of the main text. The textual information can be found in Page 13 of the main text.

For further clarification, in Page 5 of the main text, we added that the Fischer esterification protocol has been applied to couple the carboxylic acid groups with the -OH group on SiO₂ waveguide surfaces.

3. Rapid, how fast? Lifetime?

We thank Reviewer 2 for the comment. The temporal dependence of the integrated Pb^{2+} photonic sensor is investigated by the time taken for Pb^{2+} ions to bind to the crown ether-based functional layer using the X-ray Photoelectron Spectroscopy (XPS). Four sensor chips are separately exposed to 100 ppb of Pb^{2+} concentration in DI water (pH = 6.8) at exposure times of 10, 60, 120, and 180 s respectively where XPS is performed (Supplementary Fig. 1). It is observed that the signal strength increases from 10 to 60s and remain stable beyond 120 s to 180 s. It can be concluded that an equilibrium is reached between the Pb^{2+} ions and the crown ether-based functional layer is reached at 120 s. To that effect, a analyte exposure time of 120 s is maintained throughout the manuscript.

Data and description pertaining to the temporal dependence of the crown ether decorated silicon photonic Pb^{2+} detection platform is added in Supplementary Note 2 of the Supplementary. Text is added to the main text at Page 9, which points to Supplementary Note 2. The mentioned figure is reproduced below:

Supplementary Fig. 1 Sensor temporal dependence. Corresponding XPS spectrum measured from the photonic chip where the analyte comprises of 100 ppb of Pb^{2+} reference concentration in DI water. The exposure times are varied from **a**, 10, **b**, 60, **c**, 120, and **d**, 180 s. The pH of the analyte is maintained at 6.8 (see Methods).

4. *Is crown ether-based lead detection reproducible?*

We thank Reviewer 2 for the comment. Reproducibility is an imperative metric for sensors. In this revision, we have addressed the reproducibility of the Pb^{2+} detection photonic platform. The revised manuscript comprises of the measurement of Pb^{2+} at varying concentrations, pH, and in environmentally related water sources using 6 independent and separate photonic sensors ($n = 6$) for each condition.

- Analyte consisting of reference Pb^{2+} concentrations of 1, 10, 15, 80 ppb and 62 ppm in DI water with a pH of 6.8 (Table 1 of the main text)
- Analyte consisting of reference Pb^{2+} concentration of 15 ppb, at varying pH of 5, 6, 6.8, and 8, in DI water (Table 2 of the main text)
- Analyte consisting of reference Pb^{2+} concentrations of 2 ppb, with a pH of 7.26, in tap water. The reference Pb^{2+} concentration in lake water is also synthetically increased by 15 ppb to 17 ppb (Table 3 of the main text)
- Analyte consisting of reference Pb^{2+} concentrations of 5 ppb, with a pH of 7.79, in lake water. The reference Pb^{2+} concentration in lake water is also synthetically increased by 15 ppb to 20 ppb (Table 3 of the main text)
- Analyte consisting of reference Pb^{2+} concentrations of 8 ppb, with a pH of 7.91, in sea water. The reference Pb^{2+} concentration in lake water is also synthetically increased by 8 ppb to 23 ppb (Table 3 of the main text)

Table 1, 2 and 3 indicates the inferred sensor concentration $\text{Conc.}_{\text{mean}}$, standard deviation of sensor inferred concentration σ_n , and the sensor detection accuracy Err. ($\text{Conc.}_{\text{mean}}$ from reference concentrations). It is noted that towards the lower limit of detection (0.882 ppb), absolute sensor accuracies is important. On the other hand, approaching the upper limit of detection, fractional accuracies is pertinent (Supplementary Note 11). Low σ_n has been obtained in all of the abovementioned conditions implying the repeatability of the sensor.

The repeatability of crown ether decorated silicon photonic Pb^{2+} detection platform is detailed in Table 1, 2 and 3 of the main text. The mentioned tables are reproduced below:

Reference Pb^{2+} concentration in analyte (DI water, pH = 6.8)	Inferred $\text{Conc.}_{\text{mean}}/ \sigma_{\text{conc.}} (n = 6)/ \text{Err.}$
1 ppb	1.24 ppb/ 0.26 ppb/ 24.0 %
10 ppb	10.2 ppb/ 0.3 ppb/ 2.0 %
15 ppb	14.3 ppb/ 0.6 ppb/ 4.7 %

80 ppb	79.0 ppb/ 1.7 ppb/ 1.3 %
62 ppm	64.6 ppm/ 4.5 ppm/ 4.2 %

Table 1. Detection accuracy and repeatability ($\text{Conc.}_{\text{mean}}/\sigma_{\text{conc.}}$) of the crown ether decorated SiP Pb^{2+} sensor against reference Pb^{2+} concentrations, in DI water, verified by ICP-MS. The pH of the analyte is 6.8 (see Methods). $\text{Conc.}_{\text{mean}}$ refers to mean sensor inferred concentrations. $\sigma_{\text{conc.}}$ refers to standard deviation of the sensor inferred concentrations. Err. refers to the variation in $\text{Conc.}_{\text{mean}}$ from the reference concentrations. Each reference Pb^{2+} concentration was measured using six independent photonic sensors ($n = 6$).

Analyte pH (DI water, reference Pb^{2+} concentration = 15 ppb)	Inferred $\text{Conc.}_{\text{mean}}/\sigma_{\text{conc.}}$ ($n = 6$)/ Err.
pH = 5	8.2 ppb/ 0.5 ppb/ 45.3 %
pH = 6	14.7 ppb/ 0.5 ppb/ 2.0 %
pH = 6.8	15.5 ppb/ 0.7 ppb/ 3.3 %
pH = 8	14.8 ppb/ 0.6 ppb/ 1.3 %

Table 2. Detection accuracy and repeatability ($\text{Conc.}_{\text{mean}}/\sigma_{\text{conc.}}$ / Err.) of the crown ether decorated SiP Pb^{2+} sensor against analyte pH, using DI water, with reference Pb^{2+} concentrations of 15 ppb, verified by ICP-MS. $\text{Conc.}_{\text{mean}}$ refers to mean sensor inferred concentrations. $\sigma_{\text{conc.}}$ refers to standard deviation of the sensor inferred concentrations. Err. refers to the variation in $\text{Conc.}_{\text{mean}}$ from the reference concentrations. Each analyte pH was measured using six independent photonic sensors ($n = 6$).

Environmental-Related Scenarios	Inferred $\text{Conc.}_{\text{mean}}/\sigma_{\text{conc.}}$ ($n = 6$)/ Err.
Tap water, undoped; 2 ppb of Pb^{2+} , measured by ICP-MS	2.40 ppb/ 0.3 ppb/20.0 %
Lake water, undoped; 5 ppb of Pb^{2+} , measured by ICP-MS	4.77 ppb/ 0.6 ppb/ 4.6 %
Sea water, undoped; 8 ppb of Pb^{2+} ; measured by ICP-MS	8.16 ppb/ 0.4 ppb/ 2.0 %

Doped tap water, where Pb^{2+} concentration is synthetically increased by 15 ppb to 17 ppb, measured by ICP-MS	16.7 ppb/ 0.5 ppb/ 1.8 %
Doped lake water, where Pb^{2+} concentration is synthetically increased by 15 ppb to 20 ppb, measured by ICP-MS	19.5 ppb/ 0.7 ppb/ 2.5 %
Doped sea water, where Pb^{2+} concentration is synthetically increased by 15 ppb to 23 ppb, measured by ICP-MS	22.8 ppb/ 0.7 ppb/ 0.9 %

Table 3. Detection accuracy and repeatability (Conc._{mean}/ σ_{conc} / Err.) of the crown ether decorated SiP Pb^{2+} sensor in environmental-related scenarios (tap, lake, sea water), using the respective samples, where the Pb^{2+} concentrations are verified by ICP-MS. Conc._{mean} refers to mean sensor inferred concentrations. σ_{conc} refers to standard deviation of the sensor inferred concentrations. Err. refers to the variation in Conc._{mean} from the reference concentrations. The measured pH of the tap, lake and sea water are 7.26, 7.91 and 7.79 respectively.

Text relating to Table 1 is provided in Page 22, 24 of the main text.

Text relating to Table 2 is provided in Page 25 of the main text.

Text relating to Table 3 is provided in Page 29 of the main text.

4.1. *Is immobilized Pb detachable or permanent?*

We thank Reviewer 2 for the question. The immobilized Pb^{2+} can be released from the crown ether-based functional layer by subjecting the sensors to 0.1 M of nitric acid (HNO_3). This would facilitate the potential reusability of the sensors for multiples detection cycles.

The binding mechanism between the DBTDA crown ether and Pb^{2+} is facilitated through coordinate covalent bonds that are formed between the electron rich oxygen and nitrogen atoms on the crown ether, and the electron deficient Pb^{2+} . This results in the formation of a stable complex in solution^{56,57}. With the addition of H^+ ions into the aqueous environment via acidification, the protonation of the nitrogen and oxygen atoms on the crown ether occurs⁵⁴.

This leads to a reduction in the availability of active sites that form coordination bonds with Pb^{2+} . Furthermore, the introduction of positive charges (H^+) would also lead to the repelling of positively-charged Pb^{2+} . The abovementioned leads to a reduction in the binding affinity between Pb^{2+} and the crown ethers. To that effect, Pb^{2+} which are binded to the crown ethers will be released as free ions in solution.

Via the concept above, the reusability of the crown ether decorated silicon photonic Pb^{2+} detection platform is investigated. The reusability is assessed in three cases: 1.) 15 ppb Pb^{2+} reference concentration in DI water, pH = 6.8 (Fig. 6c of the main text), 2.) 80 ppb Pb^{2+} reference concentration in DI water, pH = 6.8 (Fig. 5e of the main text), 3.) 23 ppb Pb^{2+} reference concentration in sea water, pH = 7.79 (Supplementary Fig. 11c of the Supplementary) for two Pb^{2+} Detection Cycles. The data from Case 1 and 3 are replicated in Supplementary Fig. 9a and b respectively under the Detection Cycle 1. On the other hand, the data from Case 2 is replicated in Supplementary Fig. 9c under Detection Cycle 1. The reference resonant wavelength of the first detection cycle and the subsequent measured resonant wavelength upon 120s of Pb^{2+} exposure and analyte flush (defined in Page 7-10 of the main text) are termed as $\lambda_{0,1}$ and $\lambda_{s,1}$. Likewise, the two resonant photonic resonant wavelengths are defined as $\lambda_{0,2}$ and $\lambda_{s,2}$ for the second Detection Cycle. Upon the first detection cycle, where the sensor inferred concentrations (Fig. 5b of the main text) are obtained via $(\lambda_{s,1}-\lambda_{0,1})$, these devices are subjected to 0.1 M of dilute nitric acid (HNO_3), for 60 minutes to regenerate the sensors. The detailed steps are elucidated as follow, starting from the first detection cycle: 1.) The photonic sensors are first exposed to DI water to obtain $(\lambda_{0,1})$, as shown in Supplementary Fig. 9a-c. 2.) The analytes highlighted in case 1, 2, and 3 are exposed to separate photonic sensors, for 120 s. 3.) Analyte flush is performed, where the $\lambda_{s,1}$ is obtained. The sensor inferred concentrations can then be obtained from Fig. 5b of the main text via $(\lambda_{s,1}-\lambda_{0,1})$, where in Supplementary Fig. 9a-c, we show an instance of the spectral shift for the three cases highlighted above. 4.) Following, the devices in the three cases are subjected to 0.1 M of dilute nitric acid (HNO_3), for 60 minutes. The Pb^{2+} ions are released from the crown ethers as free ions and flushed away via DI water. Steps 1-3 are repeated again for Detection Cycle 2. It can be seen that $\lambda_{0,1} \approx \lambda_{0,2}$ in Supplementary Fig. 9a-c, indicating that the sensors have been regenerated through the acidification process where the Pb^{2+} ions are released from the crown ethers leading to a drop in the material index of the crown ether-based functional layer; resonant wavelength blueshift will result. Furthermore, $(\lambda_{s,2}-\lambda_{0,2}) \approx (\lambda_{s,1}-\lambda_{0,1})$, implying a close match in the sensor inferred concentrations between the two detection cycles. For further validation of the protocol, the reusability of the sensors are subjected to 6 Pb^{2+} photonic sensors ($n = 6$), for each of the three analyte cases. Supplementary Table 2 highlights the sensor inferred concentrations, $\text{Conc.}_{\text{mean}}$, sensor inferred concentration standard deviation, $\sigma_{\text{conc.}}$, and concentration detection error, Err. , where a close match between the $\text{Conc.}_{\text{mean}}$ of the first, second detection cycle and the reference Pb^{2+} concentrations (verified via ICP-MS) is obtained. The reusability of the devices are validated across different Pb^{2+} reference concentrations.

An instance Pb^{2+} photonic sensor resonant wavelength shift ($\lambda_s - \lambda_0$), indicating reusability in 1.) 15 ppb reference Pb^{2+} concentration in DI water (pH = 6.8), 2.) 80 ppb reference Pb^{2+} concentration in DI water (pH = 6.8), and 3.) 23 ppb reference Pb^{2+} concentration in sea water (pH = 7.79) have been included in Supplementary Fig. 9. The mentioned figures are reproduced below:

Supplementary Fig. 9 Demonstration of sensor reusability. The optical fringe minima measured in DI water to determine $\lambda_{0,1}$, $\lambda_{0,2}$ and after analyte flush to determine $\lambda_{s,1}$, $\lambda_{s,2}$ performed over two detection cycles at Pb^{2+} concentration of **a**, 15 ppb in DI water (pH = 6.8) **b**, 23 ppb in sea water (pH = 7.79) **c**, 80 ppb in DI water (pH =

6.8). The optical fringe minima for $\lambda_{0,1}$ and $\lambda_{s,1}$ in a-c corresponds to the spectrum measured at reference and after analyte flush in Fig. 6c of the main text, Supplementary Fig. 11c, and 5e of the main text respectively. The optical fringe minima for $\lambda_{0,2}$ is measured after sensor regeneration through exposure to 0.1 M of HNO₃ for 60 minutes. The optical fringe minima for $\lambda_{s,2}$ is measured after 120 s of Pb²⁺ exposure for each of the abovementioned tested analyte and followed by analyte flush.

Following the abovementioned protocol for sensor reuse, 6 independent photonic sensors are used to validate the process under each of the three analyte cases. Supplementary Table 2 have been added, under Supplementary Note 12, which indicates the sensor mean sensor inferred concentrations, Conc._{mean}, sensor inferred concentration standard deviation, σ_{conc} and sensor detection accuracy Err. (Conc._{mean} from reference concentrations) between two Pb²⁺ detection cycles. The table mentioned is reproduced below:

Reference concentration and analyte pH	Pb ²⁺	Inferred Conc. _{mean} / σ_{conc} . (n = 6) from the first detection	Inferred Conc. _{mean} / σ_{conc} . (n = 6) from the second detection
15 ppb in DI water, pH = 6.8		15.5 ppb/ 0.7 ppb/ 3.3 %	15.4 ppb/ 0.6 ppb/ 2.7 %
23 ppb in synthetically doped sea water; concentration of Pb ²⁺ increased by 15 ppb. pH = 7.79		22.8 ppb/ 0.7 ppb/ 0.9 %	22.6 ppb/ 0.8 ppb/ 1.7 %
80 ppb in DI water, pH = 6.8		80.9 ppb/ 2.0 ppb/ 1.1 %	82.4 ppb/ 2.9 ppb/ 3.0 %

Supplementary Table 2 Demonstration of sensor detection accuracy and reusability. Pb²⁺ concentrations are 15 ppb in DI water (pH = 6.8), 23 ppb in sea water (pH = 7.79) and 80 ppb in DI water (pH = 6.8) over two detection cycles. Conc._{mean} refers to mean sensor inferred concentrations. σ_{conc} refers to standard deviation of the sensor inferred concentrations. Err. refers to the variation in Conc._{mean} from the reference concentrations. To investigate the repeatability of the protocol, six separate sensors (*n* = 6) are tested at each concentration. The reference Pb²⁺ concentrations are verified via ICP-MS.

Contextual information with regards to Supplementary Fig. 9 and Supplementary Table 2 have been added to Page 22-23 of the main text, and Supplementary Note 12.

5. How many volumes of samples is required? Is water required filtration (filtration mandatory) to prevent any blocking in the sensor platform?

We thank Reviewer 2 for the comment. The microfluidic chamber holds 0.426 ml of analyte. The analyte is filtered through a syringe filter (0.45 μm) to prevent blockage in the sensor platform.

We have added text relating to the capacity of the sensor microfluidic chamber, as well as the filtering of the analyte through a syringe filter (0.45 μm) prior to sensor exposure in Page 10 of the main text.

6. What is the LOD?

We thank Reviewer 2 for the comment. The Pb^{2+} detection dynamic range (lower and upper limit of detection) of the integrated photonic sensor is defined as follows. The standard deviation of the sensor noise (σ_n) is found to be 0.014 nm. This is obtained from 10 repeated resonant wavelength measurements of the photonic sensor, when exposed to DI water. It is important to note that this accounts for intrinsic sensor resonance drift due to environmental thermal fluctuations, and the accuracy of the experimental setup in determining the resonant wavelength. Due to the large Pb^{2+} detection dynamic range of the sensor, it is pertinent to note that the upper and lower detection limits should be assessed by absolute accuracy and fractional accuracy respectively. To that effect, the lower limit of detection is found to be 0.882 ppb, corresponding to $(\lambda_s - \lambda_0)$ of 0.042 nm, which is three times the that of the standard deviation of the sensor noise ($3\sigma_n$)³⁷. On the other hand, the upper limit of detection is defined by the assessing the difference in $(\lambda_s - \lambda_0)$ about $\pm 10\%$ of the average resonant wavelength shift $(\lambda_s - \lambda_0)_{\text{mean}}$ of the highest tested concentration in Fig. 5b of the main text, that is still larger than $3\sigma_n$ ⁶⁰. This criteria will facilitate the confident quantification of Pb^{2+} concentration within the range. The upper limit of detection is found to be 62000 ppb. The difference in $(\lambda_s - \lambda_0) \pm 10\%$ about $(\lambda_s - \lambda_0)_{\text{mean}}$ is 0.068 nm; the value of $3\sigma_n$ is 0.042 nm. To further validate the abovementioned conditions in defining the Pb^{2+} photonic sensor lower and upper detection limits. Pb^{2+} detection near the lower limits have been performed and validated as shown in Fig. 5f and Table 1 of the main text. (1 ppb) The upper Pb^{2+} detection limit (62000 ppb) is affirmed in Fig. 5h and Table 1 of the main text where good fractional detection accuracies to the ground truth concentration values have been obtained.

The definition of the lower and upper bounds of Pb^{2+} detection pertaining to the Pb^{2+} integrated photonic sensor has been added via Supplementary Note 11 of the

Supplementary. In the main text, text has been added in Page 21, that points readers to Supplementary Note 11.

Discussion pertaining to photonic sensor Pb^{2+} lower and upper detection limit have been added in Page 21 of the main text.

The upper limit of detection (62000 ppb) pertaining to the Pb^{2+} integrated photonic sensor have been validated in Fig. 5h and Table 1 of the main text. Textual information is furnished in Page 22 of the main text. The mentioned figures and tables have is reproduced below:

Validation of the calibration curve (Fig. 5b) at reference concentrations of **h**, 62 ppm. For the detection of Pb^{2+} concentration via the sensor in Fig. 5b-h, the pH of the analyte is maintained at 6.8 (see Methods). DI water is used.

Reference Pb ²⁺ concentration in analyte (DI water, pH = 6.8)	Inferred Conc. _{mean} / $\sigma_{\text{conc.}}$ ($n = 6$)/ Err.
1 ppb	1.24 ppb/ 0.26 ppb/ 24.0 %
10 ppb	10.2 ppb/ 0.3 ppb/ 2.0 %
15 ppb	14.3 ppb/ 0.6 ppb/ 4.7 %
80 ppb	79.0 ppb/ 1.7 ppb/ 1.3 %
62 ppm	64.6 ppm/ 4.5 ppm/ 4.2 %

Table 1. Detection accuracy and repeatability (Conc._{mean}/ $\sigma_{\text{conc.}}$) of the crown ether decorated SiP Pb²⁺ sensor against reference Pb²⁺ concentrations, in DI water, verified by ICP-MS. The pH of the analyte is 6.8 (see Methods). Conc._{mean} refers to mean sensor inferred concentrations. $\sigma_{\text{conc.}}$ refers to standard deviation of the sensor inferred concentrations. Err. refers to the variation in Conc._{mean} from the reference concentrations. Each reference Pb²⁺ concentration was measured using six independent photonic sensors ($n = 6$).

6.1. It seems that the calibration curves are not linear or have a narrow of working range

We thank Reviewer 2 for the comment. The calibration curve in Fig. 5b of the main text is nonlinear. It is noted that the calibration curves of some sensors are not linear and as such, attempts to model such sensor calibration curves using a linear function will significantly restrict the detection range of the sensor. To that effect, Frankær et al. have shown that by accurately modelling the sensor calibration data to a sigmoidal curve, a significantly larger pH range can be detected, with higher precision⁶¹. In this work, the wavelength shift from reference resonance ($\lambda_s - \lambda_0$) is obtained by exposing the sensor to reference Pb²⁺ concentrations of 5, 25, 125, 625, 2625, 12625, 62625 and 262625 ppb via a cumulative testing process (see Methods). A large Pb²⁺ detection dynamic range has been experimentally verified as shown in Fig. 5f, h and Table 1 of the main text. The average wavelength shift at each tested concentration was taken, and fitted to the equation below:

$$\text{Resonance shift } (\lambda_s - \lambda_0)_{\text{average}} = \frac{a}{1 + \exp(-b \times (\text{conc.} - c))}$$

conc. is the reference Pb²⁺ concentrations (log-scale) as abovementioned. a, b, and c are fitting parameters. The shape of the curve is found to be characteristic of (i) absorption isotherms^{23,24} attributed by the binding of Pb²⁺ ions to the crown ether functional layer, and (ii) light-matter

interaction between the waveguide mode and the functional layer with Pb^{2+} binded. We have found that this function accurately models the sensor resonant shift, and the saturation of the binding sites on the crown ether decorated silicon photonic sensor.

We have added Supplementary Note 10 to the Supplementary which elucidates how the how the Pb^{2+} photonic sensor calibration data is modelled.

Text has been added in Page 21 of the main text which points to Supplementary Note 10.

7. Have you tried with a real sample of aqueous solution for lead detection? Any other interference from other compounds and ions?

We thank Reviewer 2 for the comment. In this revision, we addressed the viability of the crown ether decorated silicon photonic Pb^{2+} detection platform with real samples. Organic contaminants are found to exist in environmentally relevant water sources such as Singapore's lake²⁵ and sea water²⁶. The pH of the tap, lake and sea water are found to be 7.26, 7.91 and 7.79 respectively. Minute traces of the by-product of diphenylguanidine (DPG)²⁷ are also found to be present in Singapore's tap water. First of all, the elemental composition of the tap, lake and sea water are assessed via ICP-MS (Supplementary Note 14 of the Supplementary) and found to be 2, 5 and 8 ppb respectively. Following, the photonic sensor reference wavelength (λ_0) is first measured via the exposure of the sensor to DI water. Following, the DI water is flushed out, and the three independent photonic sensors are exposed to tap, lake and sea water separately for 120 s. After which, the above analyte is flushed out from the microfluidic chamber with DI water and the photonic sensor resonant redshift is measured again (λ_s). The ($\lambda_s - \lambda_0$) are found to be 0.081 nm for tap water (Fig. 7c of the main text), 0.169 for lake water (Fig. 7d of the main text), and 0.27 nm for sea water (Fig. 7e of the main text), corresponding to sensor inferred concentrations of 2.23, 5.06, and 8.16 ppb respectively via Fig. 5b of the main text, indicating a good match between the sensor inferred Pb^{2+} concentration and the references values. In addition, the concentration of Pb^{2+} in tap, lake and sea water are synthetically increased by 15 ppb to 17, 20 and 23 ppb, where a single measurement involving one photonic sensor for each sample is also performed. ($\lambda_s - \lambda_0$)/ sensor inferred concentration of 0.57 nm/16.9 (Supplementary 15a of the Supplementary), 0.679 nm/20.0 ppb (Supplementary 15b of the Supplementary) and 0.788 nm/ 23.1 ppb (Supplementary 15c of the Supplementary) are obtained from the tap, lake and sea water samples respectively. To investigate the photonic sensor Pb^{2+} detection accuracy and repeatability, each tap, lake and sea water sample, original and synthetically-doped are subject to 6 separate photonic sensors ($n = 6$). The mean sensor inferred concentration, $\text{Conc.}_{\text{mean}}$, standard deviation of sensor inferred concentrations, $\sigma_{\text{conc.}}$, and concentration detection error, Err. are presented in Table 3 of the main text. It is imperative to note that near the lower detection limit of the sensor, absolute sensor detection accuracy is essential. The pH neutrality of the tap, lake and sea water

implies the tendency for Pb^{2+} complexes to exist in higher proportion. The ability of the crown ether decorated silicon photonic Pb^{2+} detection platform to detect Pb^{2+} , even in the complexed state, implies the agnostic nature of the technology in detecting various forms of Pb^{2+} . This highlights one of the key strengths of the platform where laborious sample preparation is not required.

Based on the ion selectivity tests in Fig. 4, Fig. 6a-b, Supplementary Fig. 10a-i, the crown ether decorated silicon photonic Pb^{2+} detection platform is found to be only selective to Pb^{2+} .

Mention of validating the crown ether decorated silicon photonic Pb^{2+} sensing platform in environmentally-relevant situations (tap, lake and sea water) is provided in Page 6 and 30 of the main text.

Data with regards to subjecting the Pb^{2+} photonic sensor to tap, lake and sea water are provided in Fig. 7c, d, e of the main text. Data where the Pb^{2+} photonic sensor is subject to synthetically-doped (Pb^{2+} concentrations increased by 15 ppb) are indicated in Supplementary Fig. 11a, b, c of the Supplementary. Via the measurement of 6 separate and independent photonic sensors ($n = 6$) across each of the abovementioned environmentally-relevant water sources (original and synthetically-doped tap, lake and sea water sources) the information regards to mean sensor inferred concentrations $\text{Conc.}_{\text{mean}}$, standard deviation of sensor inferred concentrations σ_{conc} , and sensor detection accuracy ($\text{Conc.}_{\text{mean}}$ from reference concentration) Err. are presented in Table 3 of the main text. The textual addition are provided in Page 27-29 of the main text. The mentioned tables and figures are reproduced below:

Optical fringe minima corresponding to the detection of Pb^{2+} via the photonic sensor in **b**, tap, and **c**, lake, and **d**, sea water. **e**, XPS analysis of the crown ether functional layer when the sensor is subjected to DI, tap, lake and sea water.

Supplementary Fig. 11 Pb²⁺ sensor deployment in environmentally related situations where the concentration of tap, lake and sea water are synthetically increased by 15 ppb to a, 17, b, 20, c, 23 ppb.

The pH of the synthetically doped tap, lake and sea water are 8.26, 7.91, and 7.79 respectively.

Environmental-Related Scenarios	Inferred Conc._{mean}/ $\sigma_{\text{conc.}}$ ($n = 6$)/ Err.
Tap water, undoped; 2 ppb of Pb ²⁺ , measured by ICP-MS	2.40 ppb/ 0.3 ppb/20.0 %
Lake water, undoped; 5 ppb of Pb ²⁺ , measured by ICP-MS	4.77 ppb/ 0.6 ppb/ 4.6 %
Sea water, undoped; 8 ppb of Pb ²⁺ ; measured by ICP-MS	8.16 ppb/ 0.4 ppb/ 2.0 %
Doped tap water, where Pb ²⁺ concentration is synthetically increased by 15 ppb to 17 ppb, measured by ICP-MS	16.7 ppb/ 0.5 ppb/ 1.8 %
Doped lake water, where Pb ²⁺ concentration is synthetically increased by 15 ppb to 20 ppb, measured by ICP-MS	19.5 ppb/ 0.7 ppb/ 2.5 %
Doped sea water, where Pb ²⁺ concentration is synthetically increased by 15 ppb to 23 ppb, measured by ICP-MS	22.8 ppb/ 0.7 ppb/ 0.9 %

Table 3. Detection accuracy and repeatability (Conc._{mean}/ $\sigma_{\text{conc.}}$ / Err.) of the crown ether decorated SiP Pb²⁺ sensor in environmental-related scenarios (tap, lake, sea water), using the respective samples, where the Pb²⁺ concentrations are verified by ICP-MS. Conc._{mean} refers to mean sensor inferred concentrations. $\sigma_{\text{conc.}}$ refers to standard deviation of the sensor inferred concentrations. Err. refers to the variation in Conc._{mean} from the reference concentrations. The measured pH of the tap, lake and sea water are 7.26, 7.91 and 7.79 respectively.

Information with regards to the procurement of the environmentally-relevant water sources (tap, lake and sea water) are provided in Page 32-33 of the main text (see Methods).

8. The sensor looks very complicated in fabrication and operation.

We thank Reviewer 2 for the comment. The development of the functionalization protocol on inorganics (Fischer esterification followed by amine conjugation of crown ethers), and the photonic design is complex as elucidated in the presented manuscript. However, it is of note that these integrated photonic sensors leverages on advanced silicon manufacturing, where high-precision, large-scale manufacturing can be realized at the wafer-scale⁴⁵⁻⁴⁷. Furthermore, the crown ether functionalization process is solution based where reactants are dissolved in green solvents (i.e., water, ethanol). The abovementioned implies that the crown ether decorated silicon photonic Pb²⁺ detection platform can be implemented at scale despite its complexity in design.

9. Reorder the figure number in order correctly. It is very confusing when reading the manuscript with this complicated figure order.

We thank Reviewer 2 for the comment and we apologize for the way the figures are arranged. We have revised the figure order throughout the manuscript to facilitate ease of reading.

10. The use of crown ether amine conjugation against a wide range of relevant ions shows no binding to other ions which gives the selectivity to lead

We thank Reviewer 2 for the recognition. The selectivity to lead is facilitated by the amine conjugation of the 7,16-Dibenzyl-1,4,10,13-tetraoxa-7,16-diazacyclooctadecane (DBTDA) Crown Ether following Fischer esterification on SiO₂ waveguide surfaces.

11. Line 41, “public action dwarfs its impact” This is not clear to understand what you want to address.

We thank Reviewer 2 for the comment. We apologize for being unclear in the statement. What we meant is that current public policies have not sufficiently addressed the issue of prevalent lead poisoning in society.

We have clarified the sentence in Page 2 of the main text.

12. Line 88, Check the acronym of Environmental and Energy Law Program (EPA). Or is this a program at EPA? Clarify it

We thank the Reviewer 2 for noticing the mistake in defining "EPA". This is a mistake in defining EPA and it is not a program at EPA. We are very sorry for the mistake.

We have corrected the definition to the acronym in Page 3 of the main text.

13. Line 134, what is "SiP"? need more explanation

We thank the Reviewer 2 for the comment. "SiP" is the acronym for silicon photonic.

We have defined "SiP" as silicon photonic in Page 4 of the main text.

14. Line 153, Does "H₂O cladding" mean that the measurement is conducted in aqueous phase?

We thank the Reviewer 2 for the comment. Yes, H₂O cladding implies that the measurement is conducted in the aqueous phase. In the main text, clarification that H₂O cladding imply that the measurement is conducted in the aqueous phase is added.

In Page 7 of the main text, we have clarified that H₂O cladding refers to the measurement in the aqueous phase.

References

1. Esteban, D. *et al.* Cadmium(II) and Lead(II) Complexes with Novel Macrocyclic Receptors Derived from 1,10-Diaza-15-crown-5. *Eur J Inorg Chem* **2000**, 1445–1456 (2000).
2. Wang, D.-M. *et al.* Polymer Gels Containing Dibenzo-24-Crown-8 Ether Moieties for Removal of Cesium Ions from Aqueous Environment. *Transactions of the Materials Research Society of Japan* **44**, 217–220 (2019).
3. Cooper, T. E., Carl, D. R., Oomens, J., Steill, J. D. & Armentrout, P. B. Infrared Spectroscopy of Divalent Zinc and Cadmium Crown Ether Systems. *J Phys Chem A* **115**, 5408–5422 (2011).
4. Kumbhat, S. & Singh, U. A potassium-selective electrochemical sensor based on crown-ether functionalized self assembled monolayer. *Journal of Electroanalytical Chemistry* **809**, 31–35 (2018).
5. Ganjali, M. R., Moghimi, A. & Shamsipur, M. Beryllium-Selective Membrane Electrode Based on Benzo-9-crown-3. *Anal Chem* **70**, 5259–5263 (1998).
6. Gupta, V. K., Jain, A. K. & Kumar, P. PVC-based membranes of N,N'-dibenzyl-1,4,10,13-tetraoxa-7,16-diazacyclooctadecane as Pb(II)-selective sensor. *Sens Actuators B Chem* **120**, 259–265 (2006).
7. Abou, D. S. *et al.* Towards the stable chelation of radium for biomedical applications with an 18-membered macrocyclic ligand. *Chem. Sci.* **12**, 3733–3742 (2021).
8. Oral, I. & Abetz, V. Improved alkali metal ion capturing utilizing crown ether-based diblock copolymers in a sandwich-type complexation. *Soft Matter* **18**, 934–937 (2022).
9. Kazemi, S. Y. & Hamidi, A. S. Competitive Removal of Lead(II), Copper(II), and Cadmium(II) Ions through a Bulk Liquid Membrane Containing Macrocyclic Crown Ethers and Oleic Acid as Ion Carriers. *J Chem Eng Data* **56**, 222–229 (2011).
10. Gatto, V. J. & Gokel, G. W. Syntheses of calcium-selective, substituted diaza-crown ethers: a novel, one-step formation of bibracchial lariat ethers (BiBLES). *J Am Chem Soc* **106**, 8240–8244 (1984).
11. Vaidya, B., Porter, M. D., Utterback, M. D. & Bartsch, R. A. Selective Determination of Cadmium in Water Using a Chromogenic Crown Ether in a Mixed Micellar Solution. *Anal Chem* **69**, 2688–2693 (1997).
12. Khan, Z. *et al.* Current developments in esterification reaction: A review on process and parameters. *Journal of Industrial and Engineering Chemistry* **103**, 80–101 (2021).
13. Rothschild, E. O. Lead Poisoning — The Silent Epidemic. *New England Journal of Medicine* **283**, 704–705 (1970).
14. Dudev, T., Grauffel, C. & Lim, C. How Pb²⁺ Binds and Modulates Properties of Ca²⁺-Signaling Proteins. *Inorg Chem* **57**, 14798–14809 (2018).
15. Cory-Slechta, D. A. Legacy of Lead Exposure: Consequences for the Central Nervous System. *Otolaryngology–Head and Neck Surgery* **114**, 224–226 (1996).
16. Edwards, M. Fetal Death and Reduced Birth Rates Associated with Exposure to Lead-Contaminated Drinking Water. *Environ Sci Technol* **48**, 739–746 (2014).
17. pmc_5662044.
18. Samarghandian, S. *et al.* A systematic review of clinical and laboratory findings of lead poisoning: lessons from case reports. *Toxicol Appl Pharmacol* **429**, 115681 (2021).

19. Jurgens, B. C., Parkhurst, D. L. & Belitz, K. Assessing the Lead Solubility Potential of Untreated Groundwater of the United States. *Environ Sci Technol* **53**, 3095–3103 (2019).
20. Demir, F. & Derun, E. M. Modelling and optimization of gold mine tailings based geopolymer by using response surface method and its application in Pb²⁺ removal. *J Clean Prod* **237**, 117766 (2019).
21. M.A. Barakat M.H. Ramadan, J. N. K. & Woodcock, H. L. Equilibrium and kinetics of Pb²⁺ adsorption from aqueous solution by dendrimer/titania composites. *Desalination Water Treat* **52**, 5869–5875 (2014).
22. Sia, J. X. B. *et al.* Wafer-Scale Demonstration of Low-Loss (~0.43 dB/cm), High-Bandwidth (>38 GHz), Silicon Photonics Platform Operating at the C-Band. *IEEE Photonics J* **14**, 1–9 (2022).
23. Buttersack, C. Modeling of type IV and V sigmoidal adsorption isotherms. *Phys. Chem. Chem. Phys.* **21**, 5614–5626 (2019).
24. Altun, A. O., Bond, T., Pronk, W. & Park, H. G. Sensitive Detection of Competitive Molecular Adsorption by Surface-Enhanced Raman Spectroscopy. *Langmuir* **33**, 6999–7006 (2017).
25. Xu, Y., Luo, F., Pal, A., Gin, K. Y.-H. & Reinhard, M. Occurrence of emerging organic contaminants in a tropical urban catchment in Singapore. *Chemosphere* **83**, 963–969 (2011).
26. Basheer, C., Obbard, J. P. & Lee, H. K. Persistent organic pollutants in Singapore's coastal marine environment: Part I, seawater. Preprint at (2003).
27. Marques dos Santos, M. & Snyder, S. A. Occurrence of Polymer Additives 1,3-Diphenylguanidine (DPG), N-(1,3-Dimethylbutyl)-N'-phenyl-1,4-benzenediamine (6PPD), and Chlorinated Byproducts in Drinking Water: Contribution from Plumbing Polymer Materials. *Environ Sci Technol Lett* **10**, 885–890 (2023).
28. Nyika, J., Onyari, E., Dinka, M. O. & Mishra, S. B. A Comparison of Reproducibility of Inductively Coupled Spectrometric Techniques in Soil Metal Analyses. *Air, Soil and Water Research* **12**, 1178622119869002 (2019).
29. Douvris, C., Vaughan, T., Bussan, D., Bartzas, G. & Thomas, R. How ICP-OES changed the face of trace element analysis: Review of the global application landscape. *Science of The Total Environment* **905**, 167242 (2023).
30. Zhan, S. *et al.* Label-free fluorescent sensor for lead ion detection based on lead(II)-stabilized G-quadruplex formation. *Anal Biochem* **462**, 19–25 (2014).
31. Du, X. *et al.* A Fluorescence Sensor for Pb²⁺ Detection Based on Liquid Crystals and Aggregation-Induced Emission Luminogens. *ACS Appl Mater Interfaces* **13**, 22361–22367 (2021).
32. Marbella, L., Serli-Mitasev, B. & Basu, P. Development of a Fluorescent Pb²⁺ Sensor. *Angewandte Chemie International Edition* **48**, 3996–3998 (2009).
33. Niu, X. *et al.* A “turn-on” fluorescence sensor for Pb²⁺ detection based on graphene quantum dots and gold nanoparticles. *Sens Actuators B Chem* **255**, 1577–1581 (2018).
34. Munir, A. *et al.* Selective and simultaneous detection of Zn²⁺, Cd²⁺, Pb²⁺, Cu²⁺, Hg²⁺ and Sr²⁺ using surfactant modified electrochemical sensors. *Electrochim Acta* **323**, 134592 (2019).
35. Zhang, T. *et al.* Detection of trace Cd²⁺, Pb²⁺ and Cu²⁺ ions via porous activated carbon supported palladium nanoparticles modified electrodes using SWASV. *Mater Chem Phys* **225**, 433–442 (2019).

36. Hwang, J.-H. *et al.* Improving Electrochemical Pb²⁺ Detection Using a Vertically Aligned 2D MoS₂ Nanofilm. *Anal Chem* **91**, 11770–11777 (2019).
37. Wang, Y. *et al.* PPy-Functionalized NiFe₂O₄ Nanocomposites toward Highly Selective Pb²⁺ Electrochemical Sensing. *ACS Sustain Chem Eng* **10**, 6082–6093 (2022).
38. Liu, J. & Lu, Y. Accelerated Color Change of Gold Nanoparticles Assembled by DNazymes for Simple and Fast Colorimetric Pb²⁺ Detection. *J Am Chem Soc* **126**, 12298–12305 (2004).
39. Zhang, L. *et al.* A solid-state colorimetric fluorescence Pb²⁺-sensing scheme: mechanically-driven CsPbBr₃ nanocrystallization in glass. *Nanoscale* **12**, 8801–8808 (2020).
40. Yap, S. H. K. *et al.* An Advanced Hand-Held Microfiber-Based Sensor for Ultrasensitive Lead Ion Detection. *ACS Sens* **3**, 2506–2512 (2018).
41. PalmSens4. <https://www.palmsens.com/product/palmsens4/>.
42. Ocean ST VIS Microspectrometer. <https://www.oceaninsight.com/products/spectrometers/microspectrometer/st-series-spectrometer/ocean-st-vis-microspectrometer/>.
43. Almost 1 million people die every year due to lead poisoning, with more children suffering long-term health effects. <https://www.who.int/news/item/23-10-2022-almost-1-million-people-die-every-year-due-to-lead-poisoning--with-more-children-suffering-long-term-health-effects> (2022).
44. Perry Gottesfeld. The Environmental And Health Impacts Of Lead Battery Recycling. https://wedocs.unep.org/bitstream/handle/20.500.11822/13943/1_ECOWAS%20lead%20background%202016.pdf.
45. Sun, C. *et al.* Single-chip microprocessor that communicates directly using light. *Nature* **528**, 534–538 (2015).
46. Sun, J., Timurdogan, E., Yaacobi, A., Hosseini, E. S. & Watts, M. R. Large-scale nanophotonic phased array. *Nature* **493**, 195–199 (2013).
47. Qiang, X. *et al.* Large-scale silicon quantum photonics implementing arbitrary two-qubit processing. *Nat Photonics* **12**, 534–539 (2018).
48. Kita, D. M. *et al.* High-performance and scalable on-chip digital Fourier transform spectroscopy. *Nat Commun* **9**, 4405 (2018).
49. Sia, J. X. B. *et al.* Wafer-Scale Demonstration of Low-Loss (~0.43 dB/cm), High-Bandwidth (>38 GHz), Silicon Photonics Platform Operating at the C-Band. *IEEE Photonics J* **14**, 1–9 (2022).
50. Fahrenkopf, N. M. *et al.* The AIM Photonics MPW: A Highly Accessible Cutting Edge Technology for Rapid Prototyping of Photonic Integrated Circuits. *IEEE Journal of Selected Topics in Quantum Electronics* **25**, 1–6 (2019).
51. Siew, S. Y. *et al.* Review of Silicon Photonics Technology and Platform Development. *J. Lightwave Technol.* **39**, 4374–4389 (2021).
52. Rahim, A. *et al.* Open-Access Silicon Photonics Platforms in Europe. *IEEE Journal of Selected Topics in Quantum Electronics* **25**, 1–18 (2019).
53. Chowdhury, I. R., Chowdhury, S., Mazumder, M. A. J. & Al-Ahmed, A. Removal of lead ions (Pb²⁺) from water and wastewater: a review on the low-cost adsorbents. *Appl Water Sci* **12**, 185 (2022).
54. Golcs, Á., Vezse, P., Ádám, B. Á., Huszthy, P. & Tóth, T. Comparison in practical applications of crown ether sensor molecules containing an acridone or an acridine unit

- a study on protonation and complex formation. *J Incl Phenom Macrocycl Chem* **101**, 63–75 (2021).
55. Understanding the Lead and Copper Rule.
 56. Lundgren, R. J. & Stradiotto, M. Key Concepts in Ligand Design. in *Ligand Design in Metal Chemistry* 1–14 (John Wiley & Sons, Ltd, 2016). doi:<https://doi.org/10.1002/9781118839621.ch1>.
 57. Gupta, V. K., Jain, A. K. & Kumar, P. PVC-based membranes of N,N'-dibenzyl-1,4,10,13-tetraoxa-7,16-diazacyclooctadecane as Pb(II)-selective sensor. *Sens Actuators B Chem* **120**, 259–265 (2006).
 58. Wang, L. *et al.* Complexation mechanism of crown ethers with rubidium and cesium ions using density functional theory. *Comput Theor Chem* **1225**, 114139 (2023).
 59. Wang, Q., Wang, X. & Li, L. Density functional theory study on a fluorescent chemosensor device of aza-crown ether. *J Phys Org Chem* **27**, 546–554 (2014).
 60. T, W., H, J., A, S., P, R. & J, S. Guidance Document on the Estimation of LOD and LOQ for Measurements in the Field of Contaminants in Feed and Food. (2016) doi:10.2787/8931.
 61. Frankær, C. G. *et al.* Tuning the pKa of a pH Responsive Fluorophore and the Consequences for Calibration of Optical Sensors Based on a Single Fluorophore but Multiple Receptors. *ACS Sens* **4**, 764–773 (2019).

REVIEWERS' COMMENTS

Reviewer #1 (Remarks to the Author):

I am happy with the revision version of manuscript. It can be accepted for publication. Congratulations!

Reviewer #2 (Remarks to the Author):

The authors addressed the required information that I requested well.